# LEARNING INTERPRETABLE HIERARCHICAL DYNAMICAL SYSTEMS MODELS FROM TIME SERIES DATA

**Manuel Brenner**[1,2,*], **Elias Weber**[1,*], **Georgia Koppe**[2,3,†], **Daniel Durstewitz**[1,2,†]

[1]Dept. of Theoretical Neuroscience, Central Institute of Mental Health (CIMH)
[2]Interdisciplinary Center for Scientific Computing, Heidelberg University
[3]Hector Institute for AI in Psychiatry and Dept. of Psychiatry and Psychotherapy, CIMH
{manuel.brenner, daniel.durstewitz}@zi-mannheim.de
[*], [†]These authors contributed equally.

## ABSTRACT

In science, we are often interested in obtaining a generative model of the underlying system dynamics from observed time series. While powerful methods for dynamical systems reconstruction (DSR) exist when data come from a single domain, how to best integrate data from multiple dynamical regimes and leverage it for generalization is still an open question. This becomes particularly important when individual time series are short, and group-level information may help to fill in for gaps in single-domain data. Here we introduce a hierarchical framework that enables to harvest group-level (multi-domain) information while retaining all single-domain characteristics, and showcase it on popular DSR benchmarks, as well as on neuroscience and medical data. In addition to faithful reconstruction of all individual dynamical regimes, our unsupervised methodology discovers common low-dimensional feature spaces in which datasets with similar dynamics cluster. The features spanning these spaces were further dynamically highly interpretable, surprisingly in often linear relation to control parameters that govern the dynamics of the underlying system. Finally, we illustrate transfer learning and generalization to new parameter regimes, paving the way toward DSR foundation models.

## 1 INTRODUCTION

In scientific and medical applications, beyond mere time series forecasting, we are often interested in interpretable and mathematically tractable models that can be used to obtain mechanistic insights into the underlying system dynamics. Such models need to be *generative* in a *dynamical systems sense* (Durstewitz et al., 2023), i.e., when simulated, should produce dynamical behavior with the same *long-term statistics* as the observed system, a property that conventional time series models often lack (Fig. 29, Gilpin (2024)). This is called *dynamical systems (DS) reconstruction (DSR)*, a currently burgeoning field in scientific ML (reviewed in Durstewitz et al. (2023); Gilpin (2024)). Accounting for the fact that different time series may be sampled from fundamentally different dynamical regimes, i.e. with different state space topology and attractor structure, may furthermore endow such models with the ability to transfer knowledge and generalize (Göring et al., 2024). This is particularly important in scientific areas, where data is not only expensive to collect, hence sparse, but also subject to significant variability (Meyer-Lindenberg, 2023; Fechtelpeter et al., 2024). The challenge in these settings is to develop models that can still identify and exploit commonalities across different datasets while being flexible enough to account for individual differences in dynamical regimes.

Hierarchical models provide a natural solution by separating domain-general and domain-specific features. Such models facilitate the transfer of group-level information across individual datasets (Pan and Yang, 2010), avoiding overfitting and improving generalization to unobserved conditions with minimal data (Kirchmeyer et al., 2022). They further explicitly represent inter-domain variability, as often crucial in addressing scientific hypotheses, by unsupervised extraction of a common feature space (Zeidman et al., 2019). While hierarchical modeling is a meanwhile established approach in statistics, including time series analysis (Berliner, 1996; Hyndman et al., 2011), it has so far hardly been applied to the more challenging problem of DSR where we aim for generative models of the system dynamics that can be used for downstream analysis and simulation.

In this work, we propose such a general framework for extracting hierarchical DSR models from multi-domain time series data. Our approach combines low-dimensional, system- or subject-specific feature vectors with high-dimensional, group-level parameters to produce domain-specific DSR models, which can be trained end-to-end using state-of-the-art techniques designed for DSR (Mikhaeil et al., 2022; Hess et al., 2023). We demonstrate the effectiveness of our method in transfer and few-shot learning of DS, where generalization to previously unobserved dynamical regimes becomes possible with minimal new data. More importantly, we find that the low-dimensional subject-specific features learn to represent crucial *control parameters* of the underlying DS, rendering the subject feature spaces dynamically highly interpretable. We then demonstrate how unsupervised classification in these spaces based on *DS features* profoundly outperforms other unsupervised methods that harvest more 'conventional' time series features, providing a strong point for the DSR perspective on time series analysis.

## 2 RELATED WORK

**Dynamical Systems Reconstruction (DSR)**   DSR is a rapidly growing field in scientific ML. DSR models may be considered a special class of time series (TS) models that, beyond mere TS prediction, aim to learn surrogate models of the data-generating process from TS observations. A proper DSR model needs to preserve the *long-term temporal and geometrical properties* of the original DS (Fig. 29), i.e. its vector field topology and attractor structure, which then enables further scientific analysis (Brunton and Kutz, 2019; Durstewitz et al., 2023; Platt et al., 2023; Gilpin, 2024). A variety of DSR methods have been advanced in recent years, either based on predefined function libraries and symbolic regression, such as Sparse Identification of Nonlinear Dynamics (SINDy; Brunton et al. (2016); Kaiser et al. (2018)), based on Koopman operator theory (Azencot et al., 2020; Brunton et al., 2021; Naiman and Azencot, 2021), on universal approximators such as recurrent neural networks (RNNs; Gajamannage et al. (2023); Brenner et al. (2022); Hess et al. (2023)), reservoir computing (RC; Pathak et al. (2017); Platt et al. (2022; 2023)), switching or decomposed linear DS (Linderman et al., 2017; Mudrik et al., 2024a), neural ordinary differential equations (Neural ODEs; Chen et al. (2018); Ko et al. (2023)), or on methods that sit somewhere in between universal and domain-specific, like physics-informed neural networks (PINNs; Raissi et al. (2019)). PINNs, like library-based methods, require sufficient domain knowledge to work well in practice (Fotiadis et al., 2023; Subramanian and Mahadevan, 2023; Mouli et al., 2023). To achieve proper reconstruction of a system's long-term statistics and attractor geometry, often special, control-theoretic training techniques such as sparse or generalized teacher forcing (Brenner et al., 2022; Mikhaeil et al., 2022; Hess et al., 2023) or particular regularization terms that enforce invariant measures (Platt et al., 2023; Jiang et al., 2023; Schiff et al., 2024) are used. Models using these training techniques currently represent the state of the art in this field.

**Hierarchical Time Series and DSR Modeling**   Hierarchical models for representing multi-level dependencies or nested groups have a long history in statistics and machine learning (Laird and Ware, 1982; Goldstein, 1987; Gelman and Hill, 2006; McCulloch et al., 2011). In the time series domain, Zoeter and Heskes (2003); Bakker and Heskes (2007), for instance, introduced dynamic hierarchical models that combine individual time series with group-level dependencies. Hierarchical models have also been applied for forecasting chaotic time series (Matsumoto et al., 1998; Hyndman et al., 2011; Xu et al., 2020), or for extracting interpretable summary statistics from time series to capture inter-individual differences (Akintayo and Sarkar, 2018; Yingzhen and Mandt, 2018). Much less work exists in the direction of hierarchical DS modeling. LFADS (Pandarinath et al., 2018) and CrEIMBO (Mudrik et al., 2024b) were designed for inferring DS with neuroscience applications in mind, and can integrate observations obtained across different sessions. Roeder et al. (2019), more generally, developed a Bayesian framework using variational inference to infer Neural ODE parameters at different hierarchical levels. Yin et al. (2021); Kirchmeyer et al. (2022) used hierarchical models, which decompose dynamics into shared and environment-specific components, to enhance forecasting quality and generalization across different parameter regimes of a DS. Similarly, Bird and Williams (2019) introduced a multi-task DS model where a latent variable encodes task-specific information for sequence generation across different styles/environments. Inubushi and Goto (2020) and Guo et al. (2021) used reservoir computing for transfer learning between chaotic systems, but without explicitly modeling system-specific dynamics. Finally, Desai et al. (2022) proposed a transfer learning approach for PINNs, fine-tuning only the final layer of a pre-trained model for one-shot learning of ODEs.

In all these studies, however, only benchmark systems living in similar dynamical regimes (e.g. all exhibiting periodic orbits) were considered, often only assessing forecasting but not generative performance. This is fundamentally different from the settings with complex, chaotic, topologically diverse dynamics and real-world data we consider here, with a focus on generative and long-term performance across *different dynamical regimes*.

## 3 METHODS

### 3.1 HIERARCHICAL DSR MODEL

Assume we have observed multiple, multivariate time series $\boldsymbol{x}^{(j)}_{1\ldots T^{(j)}_{\max}}$ of lengths $T^{(j)}_{\max}$, $j = 1 \ldots S$, such as measurements from related physical systems or from multiple subjects in medical studies. While generally the individual multivariate time series may come from any type of system, in the following we will denote these as 'subjects' as our main application examples will be human data. Our main goal is to infer subject-specific DSR models, parameterized by $\boldsymbol{\theta}^{(j)}_{\text{DSR}}$, i.e. generative models of the latent dynamics underlying subject-specific observations. We approximate the dynamics by a discrete-time recursive (flow) map of the form

$$\boldsymbol{z}^{(j)}_t = \boldsymbol{F}_{\boldsymbol{\theta}^{(j)}_{\text{DSR}}}\big(\boldsymbol{z}^{(j)}_{t-1}, \boldsymbol{s}^{(j)}_t\big), \tag{1}$$

where $\boldsymbol{s}_t$ are possible external inputs (like task stimuli). Observations are related to the dynamical process $\{\boldsymbol{z}_t\}$ via some parameterized observation function

$$\hat{\boldsymbol{x}}^{(j)}_t = h_{\boldsymbol{\theta}^{(j)}_{\text{obs}}}(\boldsymbol{z}^{(j)}_t). \tag{2}$$

The *differences* between subjects are captured within a low-dimensional parameter space, represented here by learnable subject-specific features $\boldsymbol{l}^{(j)} \in \mathbb{R}^{N_{\text{feat}}}$. These are mapped onto the parameters $\boldsymbol{\theta}^{(j)}_{\text{DSR}}$ of subject-specific DSR models through a function parameterized by group-level parameters $\boldsymbol{\theta}_{\text{group}}$ common to all subjects:

$$\boldsymbol{\theta}^{(j)}_{\text{DSR}} = G_{\boldsymbol{\theta}_{\text{group}}}(\boldsymbol{l}^{(j)}). \tag{3}$$

The overall approach is illustrated in Fig. 1.

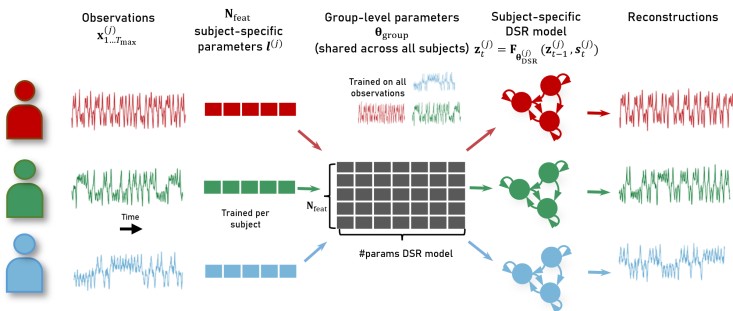

Figure 1: Illustration of the hierarchization framework.

For our experiments we chose for $\boldsymbol{F}_{\boldsymbol{\theta}^{(j)}_{\text{DSR}}}$ (Eq. 1) an RNN model introduced for DSR in Hess et al. (2023), the shallow PLRNN (shPLRNN),

$$\boldsymbol{z}^{(j)}_t = \boldsymbol{F}_{\boldsymbol{\theta}^{(j)}_{\text{DSR}}}\big(\boldsymbol{z}^{(j)}_{t-1}\big) = \boldsymbol{A}^{(j)}\boldsymbol{z}^{(j)}_{t-1} + \boldsymbol{W}^{(j)}_1\phi(\boldsymbol{W}^{(j)}_2\boldsymbol{z}^{(j)}_{t-1} + \boldsymbol{h}^{(j)}_2) + \boldsymbol{h}^{(j)}_1, \tag{4}$$

with latent states $\boldsymbol{z}^{(j)}_t \in \mathbb{R}^M$, subject-specific parameters $\boldsymbol{\theta}^{(j)}_{\text{DSR}} = \{\boldsymbol{W}^{(j)}_1 \in \mathbb{R}^{M \times L}, \boldsymbol{W}^{(j)}_2 \in \mathbb{R}^{L \times M},$ $\boldsymbol{h}^{(j)}_1 \in \mathbb{R}^M, \boldsymbol{h}^{(j)}_2 \in \mathbb{R}^L, \boldsymbol{A}^{(j)} \in \text{diag}(\mathbb{R}^M)\}$, and nonlinearity $\phi(\cdot) = \text{ReLU}(\cdot) = \max(0, \cdot)$, which makes the model piecewise-linear and thus mathematically tractable (Eisenmann et al., 2024). We also tested other models for $\boldsymbol{F}_{\boldsymbol{\theta}^{(j)}_{\text{DSR}}}$, including LSTMs (Hochreiter and Schmidhuber, 1997; Vlachas et al., 2018) and vanilla RNNs trained by GTF (Hess et al., 2023), and found that the results reported below for the shPLRNN stayed essentially the same, see Appx. A.1.6 and Table 3.

Likewise, for mapping the latent states $z_t$ to the observations $x_t$, Eq. 2, we checked different options, see Appx. A.1.6. Taking simply the identity mapping, $\hat{x}_t^{(j)} = h_{\theta_{\text{obs}}^{(j)}}(z_t^{(j)}) = z_t^{(j)}$, as in Hess et al. (2023), worked about best, presumably because a too expressive observation model can remove the burden from the DSR model to account for the actual dynamics (Hess et al., 2023).

Finally, in ablation studies we also explored various ways to parameterize the map $G_{\theta_{\text{group}}}$ in Eq. 3, including flexible and expressive functions like multilayer perceptrons (MLPs) and reparameterizations (see Appx. A.1.7, Table 4), but found that simple linear mappings often gave the best results. Specifically, for the shPLRNN in Eq. 4, the subject-level parameters $\theta_{\text{DSR}}^{(j)}$ are obtained from the feature vectors $l^{(j)} \in \mathbb{R}^{1 \times N_{\text{feat}}}$ by linearly projecting common group-level parameters $\theta_{\text{group}} = \{P_{W_1} \in \mathbb{R}^{N_{\text{feat}} \times (M \cdot L)}, P_{W_2} \in \mathbb{R}^{N_{\text{feat}} \times (L \cdot M)}, P_{h_1} \in \mathbb{R}^{N_{\text{feat}} \times M}, P_{h_2} \in \mathbb{R}^{N_{\text{feat}} \times L}, P_A \in \mathbb{R}^{N_{\text{feat}} \times M}\}$ into the subject parameter space according to:

$$A^{(j)} := \text{diag}(l^{(j)} \cdot P_A), \quad W_1^{(j)} := \text{mat}(l^{(j)} \cdot P_{W_1}, M, L), \quad W_2^{(j)} := \text{mat}(l^{(j)} \cdot P_{W_2}, L, M),$$
$$h_1^{(j)} := [l^{(j)} \cdot P_{h_1}]^T, \quad h_2^{(j)} := [l^{(j)} \cdot P_{h_2}]^T,$$

where $\text{mat}(\cdot, m, n)$ denotes the operation of reshaping a vector into an $m \times n$ matrix.

## 3.2 TRAINING PROCEDURE

All model parameters, i.e. common group-level parameters $\theta_{\text{group}}$ (matrices $P$ above) and subject-specific features $l^{(1:S)}$, are optimized simultaneously (end-to-end) by minimizing the total loss

$$\mathcal{L} = \sum_{j=1}^{S} \sum_{t=1}^{T_{\max}^{(j)}} \frac{1}{2} \log |\Sigma^{(j)}| + \frac{1}{2}(x_t^{(j)} - \hat{x}_t^{(j)})^T (\Sigma^{(j)})^{-1} (x_t^{(j)} - \hat{x}_t^{(j)}). \tag{5}$$

The total loss thus contains additional parameters $\Sigma^{(j)}$ trained jointly with $\theta_{\text{group}}$ and $l^{(1:S)}$, which are subject-specific diagonal scaling matrices introduced to deal with data that is on different scales across subjects (e.g. for observations of the Lorenz-63 system in the cyclic vs. chaotic regime). For gradient propagation, note that predictions $\hat{x}_t^{(j)}$ in Eq. 5 are linked via Eq. 2 to the latent model, Eq. 4, whose parameters $\theta_{\text{DSR}}^{(j)}$ are in turn obtained from $\theta_{\text{group}}$ and $l^{(j)}$ via Eq. 3, see Fig. 8. For the group-level matrices $P. \in \theta_{\text{group}}$, Xavier uniform initialization (Glorot and Bengio, 2010) was used and an $L_2$ norm added to the loss, see Appx. A.1 for more details.

For training, we used generalized teacher forcing (GTF) (Hess et al., 2023), a SOTA DSR training method specifically designed to address exploding gradients, which are known to occur when training DSR models on chaotic time series (Mikhaeil et al., 2022). GTF stabilizes training by linearly interpolating between RNN-generated and data-inferred control states in an optimal way (see Appx. A.1). We emphasize that GTF is only used to *train* the model, while at test time the models always run autonomously. RAdam (Liu et al., 2020) was employed as the optimizer, with significantly higher learning rate for the subject-specific feature vectors ($10^{-3}$) than for the group-level matrices ($10^{-4}$), which was crucial to prevent numerical instabilities and prioritize incorporation of subject-specific information, see Fig. 19. Across subjects, a batching strategy was employed that ensured each gradient update incorporates data from all subjects, see Appx. A.1.4.

Note that our model, although multi-level, requires only fairly few hyper-parameters: the latent size $M$ (simply taken to be equal to the number of observations in all our tests), hidden size $L$ (for which $L = 20 \times M$ is a good default), GTF parameter $\alpha$ (which could be determined automatically, see Hess et al. (2023), but here was set to defaults), and the number of features $N_{\text{feat}}$; see Appx. A.1.11.

## 4 RESULTS

### 4.1 TRANSFER LEARNING

To illustrate transfer learning, we used two popular DS benchmarks, where we sampled multivariate time series $X^{(j)}$ with varying ground-truth parameters of the underlying ODE systems: The *Lorenz-63* model (Lorenz, 1963) of atmospheric convection (Eq. 15), simulated here with 64 different parameter

combinations ('subjects') with $\rho \in \{21, 51, 81, 111\}$, $\sigma \in \{8, 9, 10, 11\}$ and $\beta \in \{1, 2, 3, 4\}$; and the spatially extended *Lorenz-96* model (Lorenz, 1996) with $N = 10$ dimensions (Eq. 17), for which 20 subjects with $F \in \{10 \cdot (j - 1) + 1\}_{j=1}^{20}$ were simulated.

The parameter ranges were chosen to cover multiple dynamical regimes, i.e. with fundamentally different attractor topologies (like limit cycles and chaos, Fig. 2). From each parameter setting, we sampled only short time series ($T_{\max} = 1000$), such that training on individual time series often led to suboptimal outcomes. Thus, leveraging group information was crucial for optimal reconstructions. As established in the field of DSR (Wood, 2010; Durstewitz et al., 2023), we evaluated the quality of reconstruction in terms of how well the reconstructed DS captured the invariant long-term temporal and geometric properties of the ground truth DS: We used a state space divergence $D_{\text{stsp}}$ to assess geometrical agreement (Koppe et al., 2019; Brenner et al., 2022) and the Hellinger distance $D_H$ on power spectra (Mikhaeil et al., 2022; Hess et al., 2023) to check temporal agreement between generated and ground truth trajectories (see Appx. A.2).

We compared the performance of our framework to three other recent methods (see Appx. A.4 for details): First, as a baseline we tested an ensemble of individual shPLRNNs, using an otherwise identical training algorithm (a kind of ablation experiment, removing specifically the 'hierarchical component'). Second, we employed LEarning Across Dynamical Systems (LEADS, Yin et al. (2021)), a framework that trains Neural ODEs for generalizing across DS environments by learning a shared dynamics model jointly with environment-specific models. Third, we trained context-informed dynamics adaptation (CoDA, Kirchmeyer et al. (2022)), an extension of LEADS where parameters of the combined and environment-specific models are drawn from a hypernetwork. As evidenced in Table 1, our hierarchical approach (hier-shPLRNN) *considerably* outperforms all other setups. In fact, competing methods were often not even able to correctly reproduce the long-term attractor dynamics (Appx. Fig. 17), while our approach successfully recovered different attractor topologies (Figs. 2 & 21). Results stayed the same when vanilla RNNs trained by GTF or LSTMs were swapped in for the shPLRNN in our framework (Appx. A.1.6 & Tab. 3), though LSTMs performed quite poorly overall.

Table 1: Performance of hierarchical and ensemble shPLRNN, CoDA (Kirchmeyer et al., 2022), and LEADS (Yin et al., 2021). Medians (across subjects) $\pm$ MAD across 10 different training runs.

| Dataset | Model | $D_{\text{stsp}}(\downarrow)$ | $D_H(\downarrow)$ | # params | # shared params |
|---|---|---|---|---|---|
| | hier-shPLRNN | **0.394 ± 0.014** | **0.097 ± 0.005** | 6912 | 6336 |
| Lorenz-63 | shPLRNN ensemble | 1.82 ± 0.16 | 0.118 ± 0.002 | 14016 | 0 |
| | CoDA | 1.4 ± 0.4 | 0.337 ± 0.005 | 97154 | 96514 |
| | LEADS | 9.7 ± 0.9 | 0.58 ± 0.04 | 8580 | 132 |
| | hier-shPLRNN | **0.580 ± 0.002** | **0.0527 ± 0.0010** | 21430 | 21130 |
| Lorenz-96 | shPLRNN ensemble | 1.291 ± 0.023 | 0.0657 ± 0.0012 | 42630 | 0 |
| | CoDA | 2.05 ± 0.07 | 0.155 ± 0.003 | 106647 | 106447 |
| | LEADS | 3.2 ± 0.5 | 0.54 ± 0.06 | 22575 | 1075 |

**Scaling and robustness** Figs. 12 & 13 show how performance scales with trajectory length $T_{\max}$ and number of subjects $S$. While performance of the hier-shPLRNN is always substantially better than when models are trained individually, this is particularly evident for short time series, $T_{\max} \leq 500$. For $T_{\max} > 500$ performance starts to plateau, but profound gains are still obtained by increasing $S$ up to 16. Surprisingly, training times until a given performance level is reached actually *decrease* with the number of subjects $S$ (Fig. 14 & Appx. A.1.9). The reason presumably is that with higher $S$, the model has access to a larger portion of the underlying system's parameter space, and can leverage this efficiently to speed up training on each single subject. Finally, robustness of discovered solutions across multiple training runs was very high, with feature vectors correlating by $r = 0.84 \pm 0.05$ on average across experiments, and as indicated by the $> 10$-fold smaller MADs in Tab. 1 for the hier-shPLRNN compared to the other approaches ($\approx 5\%$ of the resp. medians), see also Fig. 11.

## 4.2 INTERPRETABILITY

**Dynamical systems benchmarks** We assessed the ability of the hierarchical inference framework to discover interpretable structure from the Lorenz-63 (Eq. 15) and Rössler system (Eq. 16). To this end, we again sampled relatively short time series of length $T = 1000$ for 10 different values $\rho^{(j)} \in \{28 \ldots 80\}$ for the Lorenz-63, and $c^{(j)} \in \{3.8 \ldots 4.8\}$ for the Rössler, a range where its dynamics undergoes a bifurcation from limit cycle to chaos. To reflect these 1-parameter-variation

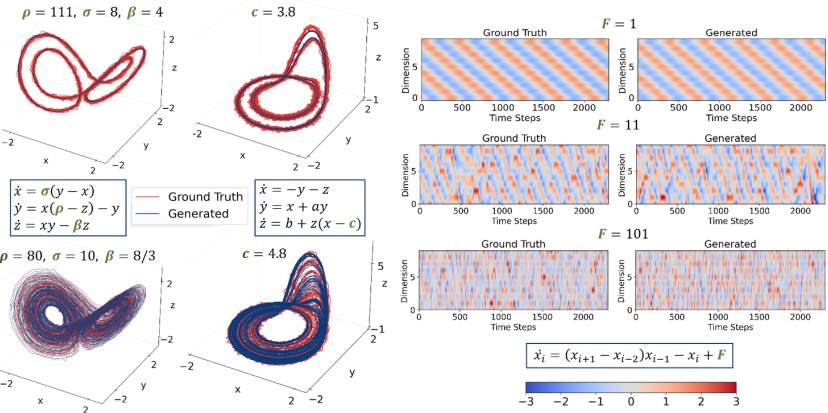

Figure 2: Example reconstructions from a hier-shPLRNN trained on short noisy observations ($T_{max} = 1000$, $5\%$ observation noise) from the Lorenz-63, Rössler and Lorenz-96 systems for different $\{\rho, c, F\}$ (settings as in Sect. 4.1 and 4.2). Shown trajectories were freely generated from a data-inferred initial state, using only subject-specific feature vectors $\boldsymbol{l}^{(j)}$ to determine the dynamical regime, and agree in their long-term temporal and geometrical structure with the ground truth (i.e., for times indefinitely beyond the short training sequences).

ground truth settings, we also chose $N_{\text{feat}} = 1$ for the length of the subject-specific parameter vectors. After training, we found that the extracted feature values $\boldsymbol{l}^{(j)}$ were highly predictive of the ground truth control parameters $\rho^{(j)}$ and $c^{(j)}$, with a clearly linear relation (Fig. 3a, see Fig. 23 for the Lorenz-96). The observation that the model automatically inferred such a linear relationship is particularly noteworthy given that the DSR model's piecewise-linear form (Eq. 4) profoundly differs from the polynomial equations defining the ground truth systems (Eqs. 15 and 16). Surprisingly, even when hier-shPLRNNs were trained with many more features ($N_{\text{feat}} = 10$) than theoretically required, a principal component analysis (PCA) on the feature vectors revealed that a dominant part of the variation was captured by the first PC ($> 85\%$ of variance), with negligible contributions beyond the third PC (Fig. 3b, blue curve).[1] The same results were obtained with a very different DSR model, Markov Neural Operators incl. a dissipativity prior (Li et al., 2022), embedded into our approach (Fig. 18). In contrast, attempts to extract low-dimensional structure from the parameters of individually trained models were unsuccessful, with variation distributed across many PCs (Fig. 3b, gray curve).

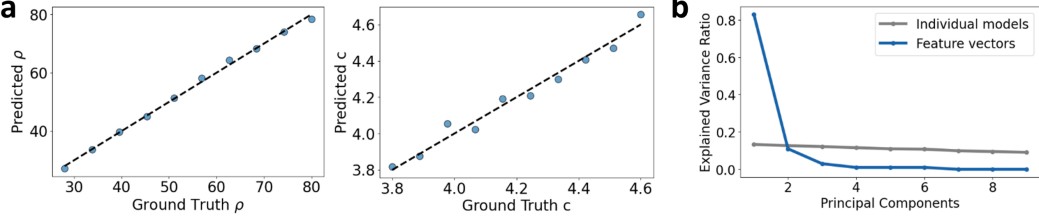

Figure 3: **a**: Analysis of one-dimensional features for hierarchical models trained on observations from the Lorenz-63 (left) and Rössler (right) system for different values of $\rho$ and $c$. **b**: Explained variance ratio of the feature space PCs for a hier-shPLRNN trained on the Lorenz-63 with $N_{\text{feat}} = 10$ vs. ratio for PCA directly on the parameters of individually trained models.

**Multidimensional parameter spaces** We next examined a scenario where all three of the ground-truth parameters $\sigma$, $\rho$ and $\beta$ of the Lorenz-63 (Eq. 15) were jointly varied across a grid of $4 \times 4 \times 4 = 64$ subjects, as in Sect. 4.1, covering both periodic and chaotic regimes. While naturally, in this case, a scalar subject-specific feature did not capture the full variation in the dataset, we found that good

---

[1]Interestingly, we observed that additional PCs reflected higher powers of the control parameters, i.e. PC2 was related to $\rho^2$ and PC3 to $\rho^3$ for the Lorenz-63.

reconstructions could already be achieved using $N_{\text{feat}} = 3$, while using $N_{\text{feat}} = 6$ features led to optimal performance. The extracted features remained highly interpretable, as illustrated in Fig. 4: $\rho$ strongly aligned with the first and $\beta$ with the second PC of the subject feature space, while variation in $\sigma$ appeared nested within steps of $\beta$ across PC2. The eight subjects on the left which do not fall into the $4 \times 4$ grid with the others, correspond to parameter combinations that put the system into a non-chaotic, cyclic regime. These subjects do not only form a distinct group in the first two PCs, but also have significant non-zero third and fourth PCs (see Fig. 22). This observation helps explain why the algorithm benefits from additional features ($N_{\text{feat}} > 3$), as these allow to represent different dynamical regimes as distinct regions in feature space, thereby further supporting its interpretability.

Fig. 28 further demonstrates that this semantic structuring of the feature spaces is observed even when the hier-shPLRNN is trained simultaneously on different dynamical regimes from *three completely different DS*, the Lorenz, the Rössler and the Chua system. The different DS are widely separated in feature space, and are further segregated according to dynamical regime within a given DS.

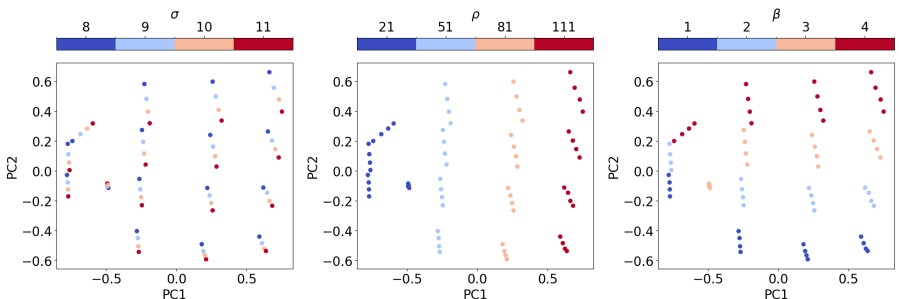

Figure 4: PCA projection of the 6d feature space for a hier-shPLRNN trained on the Lorenz-63 with variation across all 3 ground truth parameters (each dot represents one parameter combination or 'subject'). From left to right, color-coding corresponds to parameters $\sigma$, $\rho$ and $\beta$, respectively. The learned feature space is highly structured and clearly distinguishes different dynamical regimes.

**Capturing non-stationarity with time-dependent features**    One interesting use case could be if the different systems are functionally related, e.g., reflect different temporal snapshots of a non-autonomously evolving DS with time-dependent control parameter(s). In Fig. 16 we simulated such a scenario for the Lorenz-63 with $\rho(t) \in [28, 68]$ slowly drifting across time. Co-training an additional function $\tilde{l}_t^{(j)} = \text{MLP}(l^{(j)}, t)$ resulted in an explicity time-dependent DSR model which almost perfectly captured the temporal drift in $\rho(t)$ (linear $R^2 = 0.995$), confirming that our approach could in principle easily be amended to handle such situations.

**Simulated arterial pulse waves**    We then tested our approach on a much higher-dimensional ($N = 52$) example system, closer to relevant real-world scenarios, a biophysical model of arterial pulse waves (Charlton et al., 2019). The dataset consists of 4,374 simulated healthy adults aged 25–75 years, with four modalities (pressure, flow velocity, luminal area, and photoplethysmogram) across 13 body sites (see Appx. A.3 for details). Additionally, the dataset includes 32 haemodynamic parameters, such as age, heart rate, arterial size and pulse pressure amplification (Table 6), which specified the biophysical model simulations.

After training, we found that the hier-shPLRNN became an almost perfect emulator of the data, with freely generated trajectories[2] closely matching the ground truth pulse waves (Fig. 5a and Appx. Fig. 24 for further examples). This is particularly impressive given that the number of parameters of the hier-shPLRNN constituted less than $1\%$ of the total amount of training data points, and a comparatively low-dimensional (in relation to the number of biophysical parameters) feature vector ($N_{\text{feat}} = 12$) was sufficient to capture individual differences between subjects. The model's ability to achieve these reconstructions with so few parameters indicates that it extracted meaningful structure from the data, rather than merely memorizing it. Accordingly, the extracted feature vectors could predict the haemodynamic parameters via linear regression with high – and in all cases statistically

---

[2]By freely generated we mean that these are not merely forward predictions, but new trajectories drawn from the generative DSR model using only a data-inferred initial condition.

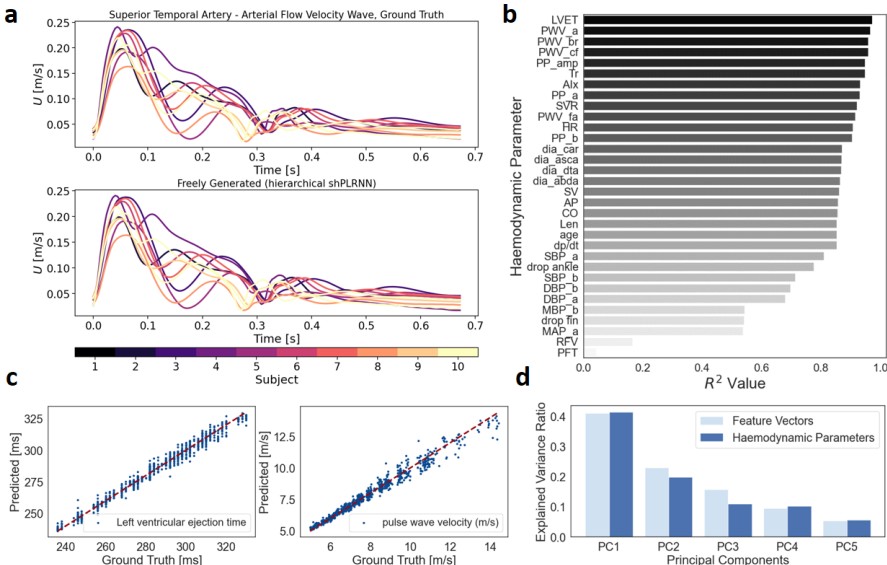

Figure 5: **a**: Ground truth and simulated (freely generated) arterial flow velocity waves for 10 representative subjects (selected by k-medoids, color-coded). **b**: $R^2$ scores for the 32 ground truth haemodynamic parameters predicted from the learned feature vectors via linear regression. **c**: Ground truth and predicted values for the left ventricular ejection time (LVET) and pulse wave velocity (PWV). **d**: Variance explained by the first 5 principal components of the learned feature space ($N_{\text{feat}} = 12$) and the 32d space spanned by the haemodynamic parameters.

significant (F-tests, $p < 0.05$) – accuracy (mean $R^2 = 0.83$, Fig. 5**b**).[3] Two parameters with a particularly strong relationship ($R^2 > 0.97$) are in Fig. 5**c**. Moreover, the explained variances of the first five principal components of the feature space agreed well with that across the haemodynamic (biophysical) parameter space (Fig. 5**d**), indicating that the learned feature space accurately captured the pattern of variation in biophysical dynamics. Again, these results are highly non-trivial as the model was trained only on the arterial pulse waves *without any knowledge of the biophysical ground truth parameters*.

## 4.3 FEW-SHOT-LEARNING

To test 'few-shot learning', by which we mean here the ability to infer new models from only small amounts of data, we first trained a hier-shPLRNN on $S = 9$ subjects simulated using the Lorenz-63 with $\rho_{\text{train}}^{(j)} = \{28, 33.8, 39.6, 45.3, 56.9, 62.7, 68.4, 74.2, 80\}$. We then generated new, short sequences $x_{1\dots T_{\text{max}}}^{\text{test}}$ with $T_{\text{max}} = 100$, using values of $\rho_{\text{test}}^{(j)}$ randomly sampled from the same interval $[28, 80]$ that also contained the training data, and fine-tuned only a scalar new feature $l_{\text{test}}^{(j)}$ on this test set. The model closely approximated (interpolated) the true $\rho_{\text{test}}^{(j)}$ values from these single short sequences (Fig. 6**a**), and was even able to *extrapolate* to new $\rho_{\text{test}}^{(j)}$ outside the training range. Fig. 6**b** exemplifies how estimates become increasingly robust for longer sequences. This result is noteworthy as training an individual shPLRNN for the Lorenz-63 is challenging even with 1000 time steps (see Table 1 & Fig. 12). Even for sequences as short as $T_{\text{max}} = 100$, the loss curves were often uni-modal and smooth around the true parameter values (Fig. 6**c**), as observed previously for training with sparse or generalized teacher forcing (Hess et al., 2023; Brenner et al., 2024). This allowed gradient descent to converge rapidly, within 6 seconds on a single 11$^{\text{th}}$ Gen 2.30 GHz Intel Core i7-11800H. It also enabled efficient use of 2$^{\text{nd}}$-order, Hessian-based methods, which converged within just 1 second. Hence, even with minimal data fast and reliable estimation of previously unseen dynamics is feasible, extending into regimes beyond the training domain.

---

[3]Further, albeit small, improvements in accuracy could be achieved by nonlinear regression via random forests (mean $R^2 = 0.87$).

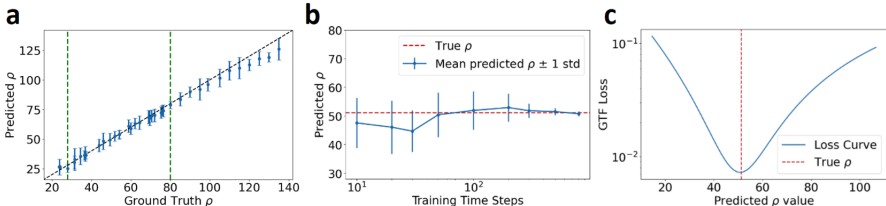

Figure 6: **a**: Predictions (by linear regression as above) for 25 $\rho_{\text{test}}^{(j)}$ values randomly drawn from the training range (green vertical lines) and several $\rho_{\text{test}}^{(j)}$ values extrapolated beyond the training range, on which only the scalar subject feature ($N_{\text{feat}} = 1$) was fine-tuned on single short test sequences ($T_{\max} = 100$) (see Appx. A.1.10 on how estimates of standard deviation were obtained). **b**: Predictions of a $\rho^{\text{test}}$ as a function of training sequence length. **c**: GTF loss for a short training sequence ($T_{\max} = 100$) across different values of the scalar subject feature mapped onto $\rho$ via linear regression. Note the uni-modal loss centered on the true $\rho^{\text{test}}$ even for this short test set.

## 4.4 EXPERIMENTS WITH HUMAN EMPIRICAL DATA

**Unsupervised extraction of DS feature spaces from clinical electroencephalogram (EEG) data** In the previous sections, we directly linked extracted feature vectors to known ground truth parameters. However, in most real-world scenarios, ground truth parameters are unavailable and we only have indirect information like class labels, making feature interpretation more challenging. To assess the performance of our hierarchical framework in such settings, we trained it on human EEG data from subjects who were either healthy or experiencing epileptic seizures (Zhang et al., 2022; Andrzejak et al., 2001), as before in a completely *unsupervised* fashion (i.e., ignoring known class labels). Using Gaussian Mixture Models (GMMs) for clustering in the extracted DS feature space, we were able to recover the true class labels with an accuracy of $92.6 \pm 1.2\%$ across ten runs, *substantially* outperforming several leading approaches for unsupervised feature extraction from time series, such as tsfresh, (Christ et al., 2018), Catch-22, (Lubba et al., 2019), ROCKET (Dempster et al., 2020), MiniRocket (Dempster et al., 2021), and attention-based convolutional autoencoders (Wang et al., 2023) (see Table 7). Fig. 7**a** shows that the two classes form distinct clusters clearly separated along the first PC of the training data, similar to the clustering of different dynamical regimes in Fig. 4, but unlike the other unsupervised techniques where classes were much less distinct. These findings confirm that clustering according to DS properties may profoundly improve on 'classical' time series feature extraction, as conjectured. Fig. 7**b** further illustrates *why* mis-classifications happened.

**Transfer learning and classification on human fMRI data** We further tested our hierarchical DSR approach on a set of short fMRI time series from 2 (hemispheres) $\times$10 distinct brain regions from 26 human subjects participating in a cognitive neuroscience experiment (Koppe et al. (2014), see Appx. A.3), of which 10 with minimal irregularities or artifacts were selected. Based on PCA (Fig. 26**c**), we determined that a feature vector dimension of $N_{\text{feat}} = 10$ was necessary to adequately capture the unique characteristics of the individual time series (Fig. 25**a**). For a left-out subject, reconstruction of the overall dynamics was feasible even when only fine-tuning the low-dimensional feature vector on unseen data (Fig. 25**b**). Since for this dataset we did not have any ground truth labels or classes, we sought to assess the robustness of feature extraction by computing the cosine similarity matrix across twenty training runs. Strong correlations ($r \approx 0.85 \pm 0.05$) indicated that feature vectors were indeed robustly inferred. Clustering the features via GMMs revealed distinct groups that aligned with visual differences observed in the fMRI time series (Fig. 26).

Finally, Fig. 27 illustrates that our hierarchical DSR approach also works for more 'classical' time series data (electricity consumption), beyond the domain of scientific applications.

## 5 CONCLUSIONS

While DSR is currently a burgeoning field, the question of how to best integrate dynamically diverse systems into a common structure, and how to harvest this for transfer learning, generalization, and classification, only recently gained traction (Yin et al., 2021; Kirchmeyer et al., 2022; Göring et al., 2024). Unlike TS models, where the interest lies mainly in forecasting or TS classification/ regression,

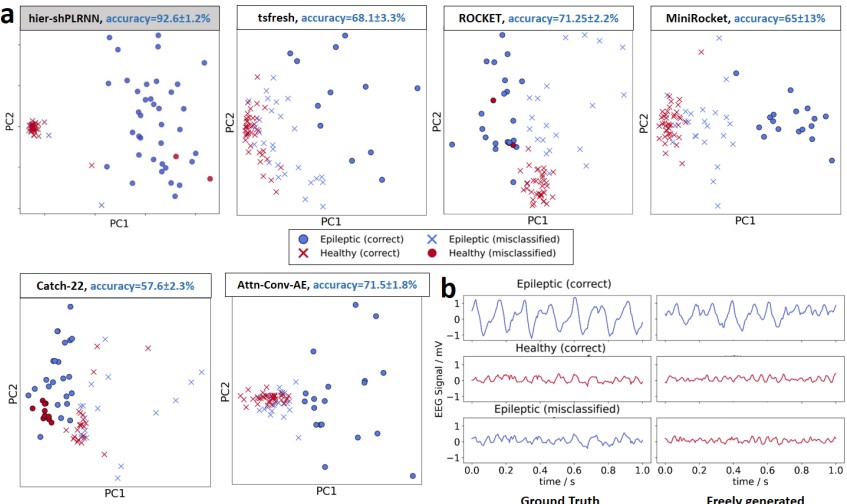

Figure 7: **a**: First two principal components of the embedding spaces of the hier-shPLRNN and other unsupervised time series feature extraction approaches from Table 7. Ground truth labels are indicated by color (blue: epileptic, red: healthy) and predicted labels are indicated by markers (circle: predicted epileptic, cross: predicted healthy). The DS feature space of the hier-shPLRNN exhibits the clearest separation of epileptic and healthy subjects (as confirmed quantitatively). **b**: Left: Ground truth data for two correctly and one incorrectly classified subjects, colored by ground truth labels. Right: Model-generated trajectories, colored by predicted labels. Note that the incorrectly classified subject is indeed dynamically much more similar to the 'healthy' than the 'epileptic' EEG.

in DSR we aim for generative models that capture invariants of the underlying system's state space. Using GTF (Hess et al., 2023) for training, we demonstrated that hierarchically structured models significantly improve DSR quality compared to non-hierarchical methods. This is a strong indication that our method was able to utilize information from all DS observed, even if located in different dynamical regimes, or if described by different systems of equations, to boost performance on each individual system. This was confirmed in transfer learning experiments, where models trained on multiple subjects generalized effectively to new conditions with minimal additional data, and did so much more convincingly than various competing methods. Identifying *dynamical regimes*, instead of just utilizing TS waveform features, is also likely to be a more principled approach to TS foundation models, since it is the underlying dynamics which determine a system's temporal properties across contexts. This idea is supported by the strong classification results on human EEG data, where subject groups were much more clearly segregated in the DS feature space than in spaces constructed by 'conventional' TS models. These DS feature spaces further turned out to be surprisingly interpretable. Extracting these dynamically most crucial features and grouping subjects along these dimensions may be of considerable interest in areas like medicine or neuroscience, and is a new application that has not been considered in the DSR field so far.

**Limitations** The highly interpretable nature of the subject-level feature spaces is surprising. It appears that our method automatically infers critical control parameters of a system that best differentiate between dynamical regimes (by controlling the underlying bifurcation structure), but it is currently unclear what the mathematical basis for this might be. Another open issue is how DS could best be integrated into the same model that are related but live on widely different time scales, e.g. brain signals over minutes vs. behavioral monitoring across weeks. While Fig. 28 and the empirical examples confirm our approach provides a means to examine how diverse systems are dynamically related, this currently is limited to DS evolving on similar time scales. In Fig. 16 we illustrated how the hier-shPLRNN could be expanded to capture non-stationarity and drivers in non-autonomous DS in an interpretable way; yet, for specific physical or biological processes this may need to be hooked up with more domain knowledge to establish a mechanistic link to the underlying physics/ biology. A useful extension thus could be the incorporation of prior physical knowledge as in SINDy or PINNs. While the latter is no principle obstacle, as shown here for Markov Neural Operators, the tricky part may be to formulate priors that work across all observed systems and domains.

## REPRODUCIBILITY STATEMENT

The code for this paper is publicly available at https://github.com/DurstewitzLab/HierarchicalDSR.

## ACKNOWLEDGEMENTS

This work was funded by the European Union's Horizon 2020 programme under grant agreement 945263 (IMMERSE), by the German Ministry for Education & Research (BMBF) within the FE-DORA (01EQ2403F) consortium, by the Federal Ministry of Science, Education, and Culture (MWK) of the state of Baden-Württemberg within the AI Health Innovation Cluster Initiative and living lab (grant number 31-7547.223-7/3/2), by the German Research Foundation (DFG) within the collaborative research center TRR-265 (project A06 & B08) and by the Hector-II foundation.

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

# A METHODOLOGICAL DETAILS

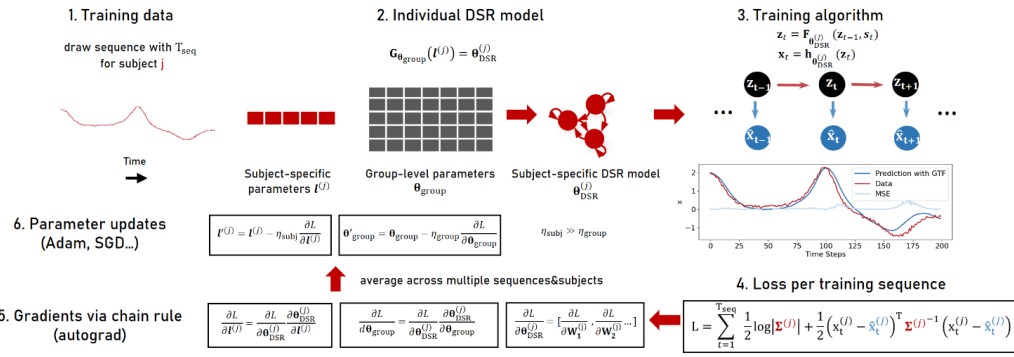

Figure 8: Illustration of the training algorithm for one example training sequence.

## A.1 MODEL TRAINING

### A.1.1 GENERALIZED TEACHER FORCING (GTF)

GTF interpolates between forward-propagated latent states $z_t$ of the DSR model and states $\bar{z}_t = h^{-1}(x_t)$ inferred from the observations according to:

$$\tilde{z}_t := (1 - \alpha)z_t + \alpha \bar{z}_t, \qquad (6)$$

with $0 \le \alpha \le 1$, which may be set by line search or adaptively (automatically) adjusted throughout training (here we simply picked reasonable defaults based on Fig. S3 in Hess et al. (2023)). Hess et al. (2023) prove that an optimal choice for $\alpha$ leads to bounded gradients as sequence length $T$ goes to infinity. An extension of GTF to multimodal and non-Gaussian data (multimodal teacher forcing, MTF) has recently been proposed (Brenner et al., 2024). MTF can be naturally incorporated into our hierarchical framework and hence is an interesting avenue for further applications.

### A.1.2 DIFFERENTIAL LEARNING RATES

For training we used the RAdam (Liu et al., 2020) optimizer. To ensure successful training, we found it crucial to assign a significantly higher learning rate to the subject-specific feature vectors ($10^{-3}$) than to the group-level matrices ($10^{-4}$). This helped prevent numerical instabilities in the group-level matrices, which occurred with higher learning rates. It also prioritizes the incorporation of subject-specific information through the feature vectors, see Fig. 19.

### A.1.3 INITIALIZATION

We used Xavier uniform initialization (Glorot and Bengio, 2010) for the group-level matrices, which is designed to keep the variance of the outputs $n_{out}$ of a layer roughly equal to the variance of its inputs $n_{in}$ by drawing weights from a uniform distribution in the interval $[-a, a]$, where $a = \sqrt{\frac{6}{n_{in} + n_{out}}}$. We reduced the weights further by a factor of $0.1$ to stabilize training. Additionally, we applied $L_2$ regularization to the group-level matrices to further enhance stability.

### A.1.4 BATCHING

We employed a batching strategy that subsamples training sequences $x_{\tau:\tau+T_{seq}^{(j)}-1}$ of length $T_{seq}^{(j)}$ from the total time series of length $T_{max}^{(j)}$, starting at a randomly drawn position $\tau \in \{1, T_{max}^{(j)} - T_{seq}^{(j)} + 1\}$. Given the complexity of the datasets considered here, an optimal data loading strategy for stochastic optimization is not obvious a priori. We hence investigated how different batch sizes impact the reconstruction performance of models trained on the 64-subject Lorenz-63 dataset (see main text). The blue curve in Fig. 9 illustrates how performance varies across a range of batch sizes when training with 50 batches per epoch, corresponding to a fixed number of gradient updates per epoch. In this

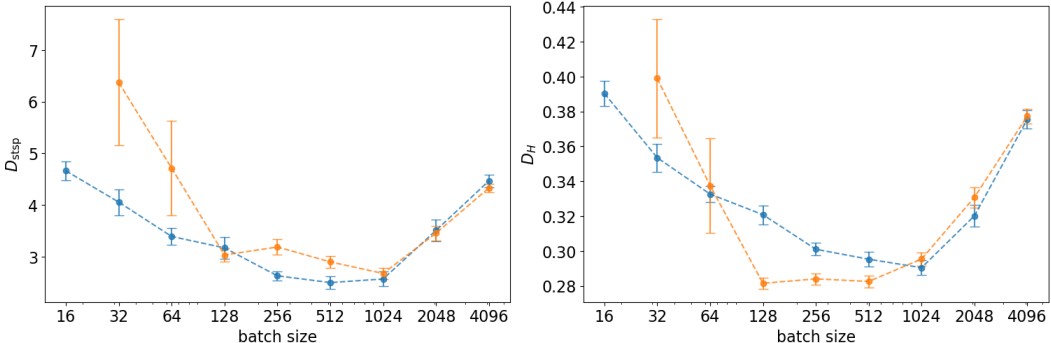

Figure 9: Mean reconstruction performance for different batch sizes (error bars = standard deviation). The blue curve corresponds to training with a fixed number of gradient steps per epoch, while the orange curve represents training with a fixed amount of data per epoch. Some models with smaller batch sizes failed to train entirely, leading to high variance for batch sizes 32 and 64 and the exclusion of batch size 16.

setup, models trained with smaller batch sizes naturally see less data per epoch than those with larger batches. To ensure that the observed effects are not merely due to reduced data exposure with smaller batch sizes, we repeated the experiment using the same batch sizes, but ensuring the entire dataset was seen in each epoch (orange curve). Our findings suggest that smaller batch sizes do not contain enough statistical information across all subjects, leading to higher variance in gradient updates and, in some cases, failure of training. For larger batch sizes, the performance difference was marginal.

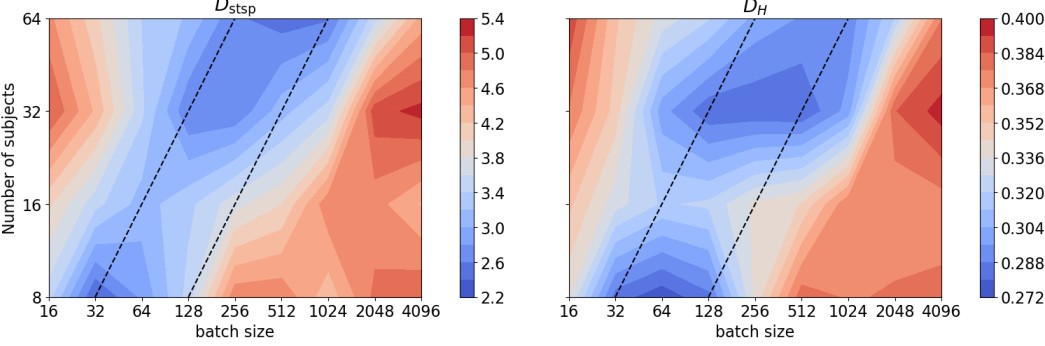

Figure 10: Reconstruction performance for different batch sizes and numbers of subjects. The optimal batch size ranges approximately between 4–16 × the number of subjects, as highlighted by the dashed black lines.

The previous experiments were conducted with a fixed number of subjects. However, since we aim to incorporate information from all subjects within each batch, we extended the analysis to examine how the optimal batch size scales with different numbers of subjects. To do this, we randomly sampled smaller subsets from the dataset and repeated the experiment. Fig. 10 gives a heat map of the results, showing that the optimal batch size falls between 4 to 16 times the number of subjects in the dataset. For datasets with a large number of subjects, this range becomes computationally infeasible, so we instead sample batch sequences from a random subset of subjects on each iteration.

### A.1.5   ROBUSTNESS

As evident from Tab. 1, repeated training runs with the hier-shPLRNN deliver highly consistent results. Fig. 11**b** dissects this further by plotting the median performance (across the $S$ subjects) for all ten training runs separately, which exhibits only little variation. We further quantified the robustness of the learned feature spaces by first calculating the pairwise cosine similarities between subject feature vectors in any given run, and then the correlations among those across ten different training

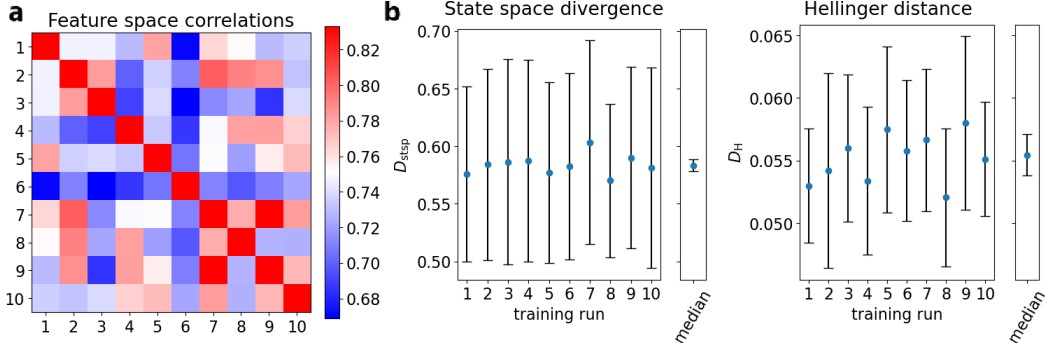

Figure 11: **a**: Cross-correlation matrix of cosine similarities between the $S = 20$ feature vectors for ten Lorenz-96 training runs. **b**: Mean performance measures across all 20 subjects for ten Lorenz-96 training runs.

runs (yielding the cross-correlation matrix in Fig. 11**a** for the Lorenz-96 benchmark). Average cross-correlations for all the different settings studied here are in Tab. 2.

Table 2: Mean ± SD of feature space correlations for all experiments.

| Experiment | Lorenz-63 | Lorenz-96 | Pulse Wave | EEG | fMRI |
|---|---|---|---|---|---|
| Correlation | $0.90 \pm 0.03$ | $0.75 \pm 0.05$ | $0.88 \pm 0.04$ | $0.81 \pm 0.03$ | $0.85 \pm 0.05$ |

### A.1.6 OTHER ARCHITECTURES TESTED

Besides the shPLRNN for which results are reported in the main text, we repeated the Lorenz-63 experiment in sect. 4.1 with the following RNN architectures trained by GTF (Hess et al., 2023): First, a simple 'vanilla RNN',

$$\boldsymbol{z}_t = \tanh(\boldsymbol{W}\boldsymbol{z}_{t-1} + \boldsymbol{b}), \tag{7}$$

and, second, an LSTM

$$
\begin{aligned}
\boldsymbol{f}_t &= \sigma\left(\boldsymbol{W}_f \boldsymbol{x}_{t-1} + \boldsymbol{U}_f \boldsymbol{z}_{t-1} + \boldsymbol{b}_f\right) \\
\boldsymbol{i}_t &= \sigma\left(\boldsymbol{W}_i \boldsymbol{x}_{t-1} + \boldsymbol{U}_i \boldsymbol{z}_{t-1} + \boldsymbol{b}_i\right) \\
\boldsymbol{o}_t &= \sigma\left(\boldsymbol{W}_o \boldsymbol{x}_{t-1} + \boldsymbol{U}_o \boldsymbol{z}_{t-1} + \boldsymbol{b}_o\right) \\
\tilde{\boldsymbol{c}}_t &= \tanh\left(\boldsymbol{W}_f \boldsymbol{x}_{t-1} + \boldsymbol{U}_f \boldsymbol{z}_{t-1} + \boldsymbol{b}_f\right) \\
\boldsymbol{c}_t &= \boldsymbol{f}_t \odot c_{t-1} + \boldsymbol{i} \odot \tilde{\boldsymbol{c}}_t \\
\boldsymbol{z}_t &= \boldsymbol{o}_t \odot \tanh\left(\boldsymbol{c}_t\right)
\end{aligned}
\tag{8}
$$

where $\odot$ denotes element-wise multiplication. Since both architectures use sigmoidal activation functions, we employ a linear observation model, shared across subjects, to scale the latent states back to the observation range.

Results in Tab. 3 indicate that while both these RNNs also profit from the hierarchisation, performance for the LSTM is considerably worse than for the shPLRNN and vanilla RNN (potentially because due to its specialized architecture the LSTM gains less from the GTF-based training; for DSR, the training technique is actually more crucial than the architecture, Hess et al. (2023)).

Additionally, we tested how using a linear observation model, i.e. $\hat{\boldsymbol{x}}_t^{(j)} = h_{\boldsymbol{\theta}_{\text{obs}}^{(j)}}(\boldsymbol{z}_t^{(j)}) = \boldsymbol{B}^{(j)} \boldsymbol{z}_t^{(j)}$, influences hier-shPLRNN performance. We tested both, using a shared observation matrix across all subjects ($\boldsymbol{B}^{(j)} = \boldsymbol{B} \ \forall j$), as well as subject-specific observation matrices constructed from the respective feature vectors ($\boldsymbol{B}^{(j)} = \text{mat}(\boldsymbol{l}^{(j)} \cdot \boldsymbol{P}_B, N, M)$). Results in Tab 3 indicate that both variants of a linear observation model degrade performance slightly.

Table 3: Median ($\pm$ MAD) reconstruction performance for the two alternative RNN architectures, and for two variants of linear observation models for the shPLRNN, on the Lorenz-63 benchmark from Sect. 4.1.

| Architecture | | $D_{\text{stsp}}$ | $D_H$ |
|---|---|---|---|
| Vanilla | ensemble | $1.72 \pm 0.24$ | $0.296 \pm 0.008$ |
| | hierarchical | $0.57 \pm 0.09$ | $0.199 \pm 0.010$ |
| LSTM | ensemble | $2.7 \pm 0.3$ | $0.76 \pm 0.017$ |
| | hierarchical | $2.3 \pm 0.3$ | $0.37 \pm 0.06$ |
| hier-shPLRNN + shared obs. model | | $0.55 \pm 0.26$ | $0.087 \pm 0.012$ |
| hier-shPLRNN + hierarchical obs. model | | $0.54 \pm 0.18$ | $0.087 \pm 0.010$ |

### A.1.7 HIERARCHIZATION SCHEME

As discussed in Sect. 3, our hierarchization framework is flexible, allowing for different forms of the group-level function $G_{\boldsymbol{\theta}_{\text{group}}}$ that maps subject-specific feature vectors to model weights. Here we discuss different choices for parameterizing $G_{\boldsymbol{\theta}_{\text{group}}}$ and points to consider in this context. The naive approach, employing direct linear projections, results in over-parameterization. For example, the weight matrices $\boldsymbol{W}_1$ and $\boldsymbol{W}_2$ of the shPLRNN have $M \times L$ degrees of freedom, but when parameterized with feature vectors through $\boldsymbol{W}_1^{(j)} := \text{mat}(\boldsymbol{l}^{(j)} \cdot \boldsymbol{P}_{W_1}, M, L)$, the number of trainable parameters expands to $N_{\text{feat}} \times (M \times L + 1)$, which with $N_{\text{feat}} > 1$ exceeds the $M \times L$ available entries in $\boldsymbol{W}_1$.

To address this, we tested using outer product forms, reducing parameter load by constructing all subject-specific weight matrices from two matrices $\boldsymbol{P}_{W_1} \in \mathbb{R}^{M \times N_{\text{feat}}}$ and $\boldsymbol{Q}_{W_1} \in \mathbb{R}^{N_{\text{feat}} \times L}$ via:

$$\boldsymbol{W}_1^{(j)} = \boldsymbol{P}_{W_1} \text{diag}(\boldsymbol{l}^{(j)}) \boldsymbol{Q}_{W_1} \tag{9}$$

(analogously for $\boldsymbol{W}_2$), while the diagonal of $\boldsymbol{A}$ and vectors $\boldsymbol{h}_1, \boldsymbol{h}_2$ were specified directly through linear maps. This reduces the number of parameters to $N_{\text{feat}} \times (M + L + 1)$. In the cases where $N_{\text{feat}} < M$, such that the matrices would not be full rank, we applied an extra step to map the feature vector to an auxiliary vector $\mathbb{R}^M \ni \boldsymbol{a}^{(j)} = \boldsymbol{R}\boldsymbol{l}^{(j)}$ via the shared[4] matrix $\boldsymbol{R} \in \mathbb{R}^{M \times N_{\text{feat}}}$, and subsequently mapped to the weights $\boldsymbol{W}_1^{(j)} = \boldsymbol{P}_{W_1} \text{diag}(\boldsymbol{a}^{(j)}) \boldsymbol{Q}_{W_1}$ (analogously for $\boldsymbol{W}_2$). We also tested using an MLP for mapping the feature vectors onto the subject-specific DSR model weights, with MLP parameters shared among subjects:

$$\boldsymbol{\theta}_{\text{DSR}}^{(j)} = \text{MLP}(\boldsymbol{l}^{(j)}) \tag{10}$$

The MLP mapped onto a flat vector, which was reshaped into the respective shPLRNN weight matrices.

Both of these schemes did, however, not significantly improve reconstruction results on the Lorenz-63 dataset (see Sect. 4.1) when compared to the naive linear approach according to one-sided t-tests, see Tab. 4.

Table 4: Median state space divergence and Hellinger distance for different parameterizations of the hierarchization scheme, for 10 models each trained on the 64 subjects created via the Lorenz-63 system. The MLP parameterization was clearly inferior, while the linear and outer product approaches were statistically indistinguishable.

| Scheme | # params | $D_{\text{stsp}}$ | | | $D_H$ | | |
|---|---|---|---|---|---|---|---|
| | | Value ($\downarrow$) | t(18) | p | Value ($\downarrow$) | t(18) | p |
| Naive linear | 6912 | $0.394 \pm 0.014$ | | | $0.097 \pm 0.005$ | | |
| Outer product | 3348 | $0.36 \pm 0.03$ | $0.99$ | $0.84$ | $0.0986 \pm 0.0025$ | $-0.63$ | $0.27$ |
| MLP | 36713 | $0.85 \pm 0.09$ | $-7.55$ | $0.0$ | $0.111 \pm 0.004$ | $-5.54$ | $0.0$ |

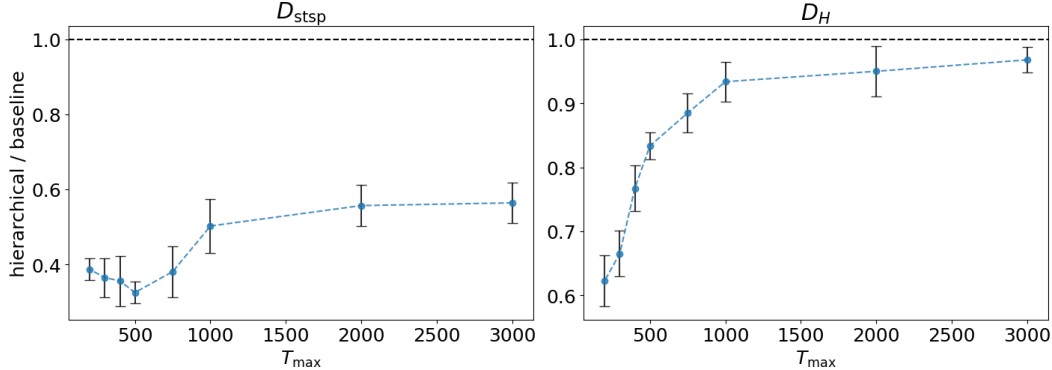

Figure 12: Relative performance (hierarchical/baseline) as a function of time series length for the 64-subject Lorenz-63 dataset (lower = better).

### A.1.8 PERFORMANCE AS A FUNCTION OF DATA SIZE

To further evaluate how performance scales with the number of subjects and available time series length, we conducted additional experiments on the Lorenz-63 dataset as in Sect. 4.1, but with a varying number of time steps for each subject/system, where 1000 training time steps corresponds to the setting in Sect. 4.1. Fig. 12 shows the ratio between the hierarchical model's performance and the baseline performance (i.e., training an ensemble of individual shPLRNNs, with values below 1 indicating superior performance of the hierarchical model). Even for long training sequences the hierarchical model consistently outperforms the baseline, particularly in terms of state-space divergence.

We furthermore evaluate how the performance depends on the number of subjects $S$ for different values of time series length $T_{max}$, using at least two different dynamical regimes (limit cycle and chaos) for the Lorenz-63 (hence starting at $S = 2$). Results are in Fig. 13, illustrating that performance gains are substantial up to $S = 16$ for this dataset.

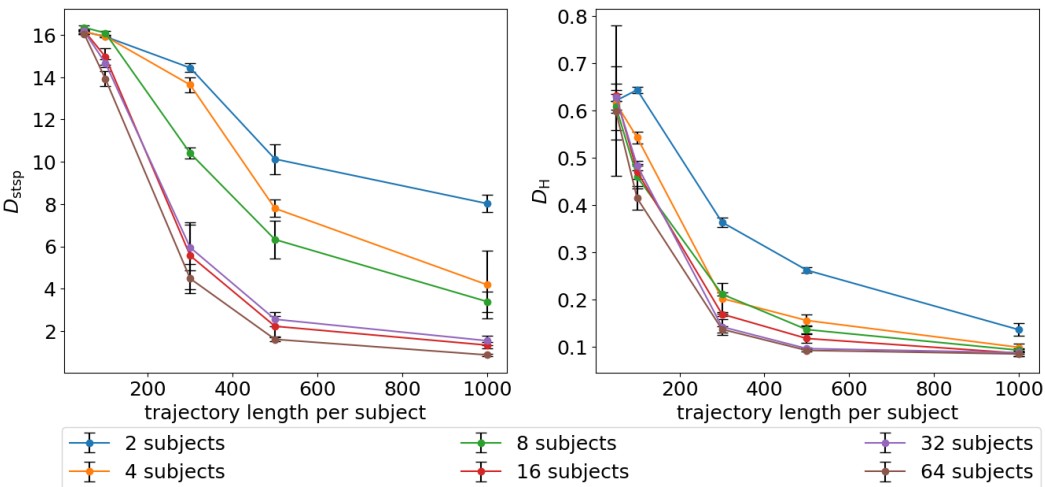

Figure 13: Dependence of model performance on the number of subjects $S$ and trajectory length $T_{max}$.

---

[4] To avoid increasing the number of free parameters, $\boldsymbol{R}$ could also be chosen to be random and constant.

### A.1.9 Computational complexity

Qua construction, *inference* times (i.e., forward-iterating the full model from some initial condition) should scale linearly with number of subjects $S$ and number of features $N_{\text{feat}}$, which Fig. 14**b-c** confirms they do (note that due to construction of the subject-specific model matrices $\boldsymbol{\theta}_{\text{DSR}}^{(j)}$, initially setup times for the hier-shPLRNN are slightly larger than for an ensemble of shPLRNNs, but this difference is negligible in practice for realistic settings with $N_{\text{feat}} << S$ and more than made up for by the fast split-second fine-tuning of the hier-shPLRNN once trained).

More importantly, as shown in Fig. 14**a**, actual training times for the hier-shPLRNN until a pre-set performance criterion was reached ($D_{\text{stsp}} \leq 1.0$), actually *decrease* with the number of subjects $S$ (note that all training here was intentionally performed on a single CPU core, hence there was no parallelization here). The reason for this presumably is that incorporation of more subjects, and hence sampling of a larger portion of the underlying systems' parameter space, allows the model to reach the criterion and reconstruct each subject's dynamics much more quickly. This further illustrates that the hier-shPLRNN leverages and transfers information from multiple subjects highly efficiently.

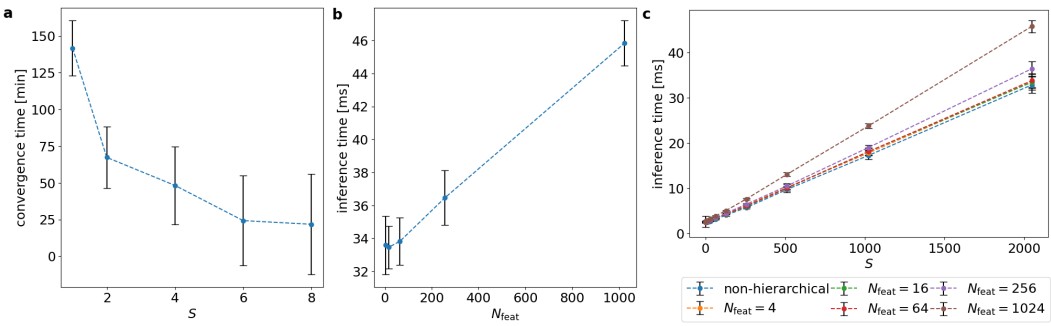

Figure 14: **a**: Wall clock time of model convergence (i.e. first passage time for median $D_{\text{stsp}} \leq 1$) for models trained with different $S$ (using the Lorenz-63 system with $\rho \in \{28, 34, 40, 46, 52, 58, 64, 70, 76, 82\}$). For all values of $S$, the total number of data points was held constant at $20,000$ time steps, uniformly divided among all subjects. Error bars = SEM across ten runs, sampling $D_{\text{stsp}}$ every 50 epochs. All models were trained on a single core of an Intel Xeon Gold 6254. **b**: Wall clock time of model inference ($T_{\text{seq}} = 30$, $M = 3$, $L = 50$) for different $N_{\text{feat}}$ at $S = 2048$. **c**: Wall clock time of model inference for different $S$ and $N_{\text{feat}}$.

### A.1.10 Uncertainty Estimates

To obtain the uncertainty estimates (error bars) in Figs.6**b** and **c**, we first drew a long trajectory with 10000 time steps from the ground truth system using $\rho^{(S+1)}$. We then randomly sampled multiple training trajectories of length $T_{\text{max}}$, inferred the feature vectors $\hat{\boldsymbol{l}}^{(S+1)}$ for each of those individually, and calculated the standard deviation of the resulting $\hat{\rho}^{(S+1)}$ estimates. As shown in Fig. 15, the estimates sharpen as longer training sequences are used.

### A.1.11 Hyperparameters

In Tab. 5 we report the hyperparameters used for training the shPLRNN models on the datasets in Sect. 4.1. These include the latent dimension $M$ (taken to be equivalent to the dimension of the observed data here, thus fixed), the hidden dimension $L$, the GTF interpolation factor $\alpha$, and the number of features $N_{\text{feat}}$. As recommended in Hess et al. (2023), $\alpha$ decays exponentially from $\alpha_{\text{start}}$ at the beginning to $\alpha_{\text{end}}$ at the end of training, where our values are based on Fig. S3 in that paper and not further fine-tuned (nor adaptively determined). The only parameters determined by grid search were thus $N_{\text{feat}}$ and $L$.

## A.2 Evaluation Measures

**State Space Divergence** $D_{\text{stsp}}$ is the Kullback-Leibler divergence between generated, $p_{\text{gen}}(x)$, and ground truth, $p_{\text{true}}(x)$, trajectory point distributions in state space (hence a measure of geometrical

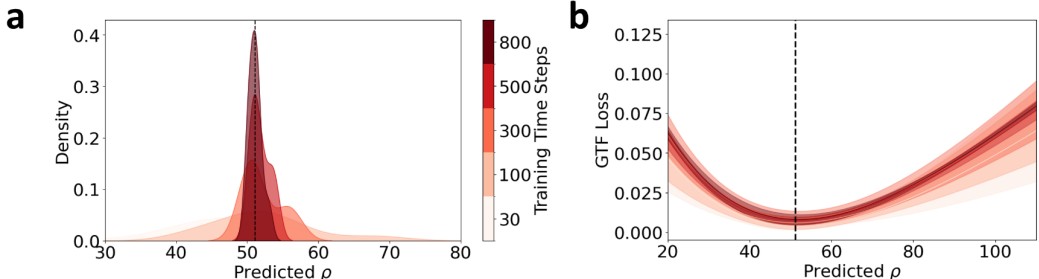

Figure 15: **a**: Distributions $p(\hat{\rho}^{(j)} \mid \boldsymbol{x}^{(j)}_{1\ldots T_{\max}})$ (approximated using kernel density estimates), estimated from 10 time series each, with different lengths $T_{\max}$ (color-coded) for the example $\rho^{(j)}_{\text{test}}$ from Fig. 6. **b**: Corresponding GTF loss curves for different training sequence lengths, illustrating that longer sequences/more training data lead to smaller standard deviation over loss curves.

Table 5: Hyperparameters for our models from Sect. 4.1.

| Benchmark | Model | $N_{\text{feat}}$ | $M$ | $L$ | $\alpha_{\text{start}}$ | $\alpha_{\text{end}}$ |
|---|---|---|---|---|---|---|
| Lorenz-63 | shPLRNN ensemble | - | 3 | 30 | 0.2 | 0.02 |
| | hier-shPLRNN | 6 | 3 | 150 | 0.2 | 0.02 |
| Lorenz-96 | shPLRNN ensemble | - | 10 | 100 | 0.2 | 0.02 |
| | hier-shPLRNN | 5 | 10 | 200 | 0.2 | 0.02 |

similarity):

$$D_{\text{stsp}} = KL(p_{\text{gen}} \| p_{\text{true}}) = \int p_{\text{gen}}(x) \log \frac{p_{\text{gen}}(x)}{p_{\text{true}}(x)} \mathrm{d}x \tag{11}$$

In lower dimensional state spaces this may be approximated by discrete binning: Let $m$ be the number of bins in each dimension and $\hat{p}^{(k)}$ the number of points in the $k$-th bin, then with a total number of bins $K = m^d$, an estimate of the state space divergence is given by

$$D_{\text{stsp}} \simeq \sum_{k=1}^{K} \hat{p}^{(k)}_{\text{gen}}(x) \log \frac{\hat{p}^{(k)}_{\text{gen}}(x)}{\hat{p}^{(k)}_{\text{true}}(x)} \tag{12}$$

Since the total number of bins scales exponentially with the observation dimension $N$, for experiments with the Lorenz-96 system, we used $m = 5$ per dimension to ensure the approach remained computationally feasible. For very high-dimensional spaces, Brenner et al. (2022); Mikhaeil et al. (2022) suggest a GMM-based estimate with Gaussians centered on the trajectory points (which was not required here, however). For the values in Table 1, we sampled single time series with $10,000$ time steps from both the ground truth system (serving as the test set) and from each trained model, inferring only the initial state from the first time step of the test set, and cutting off transients ($1,000$ time steps) to focus on the long term attractor behavior.

**Average Hellinger Distance** To assess long-term temporal agreement, we calculated the average Hellinger distance $D_H$ between the power spectra of true and generated time series (Mikhaeil et al., 2022). Let $f_i(\omega)$ be the power spectrum for the $i$-th dimension (time series variable), then the Hellinger distance averaged across all dimensions is defined as

$$D_H = \frac{1}{d} \sum_{i=1}^{d} \sqrt{1 - \int \sqrt{f_i^{(\text{true})}(\omega) f_i^{(\text{gen})}(\omega)} \mathrm{d}\omega} \tag{13}$$

The respective power spectra are estimated by the Fast Fourier Transform (FFT), which leads to the integral approximation

$$D_H \simeq \frac{1}{d} \sum_{i=1}^{d} \sqrt{1 - \sum_j \sqrt{f_i^{(\text{true})}(\omega_j) f_i^{(\text{gen})}(\omega_j)}} \tag{14}$$

Test set and model-generated trajectories were computed as for $D_{\text{stsp}}$.

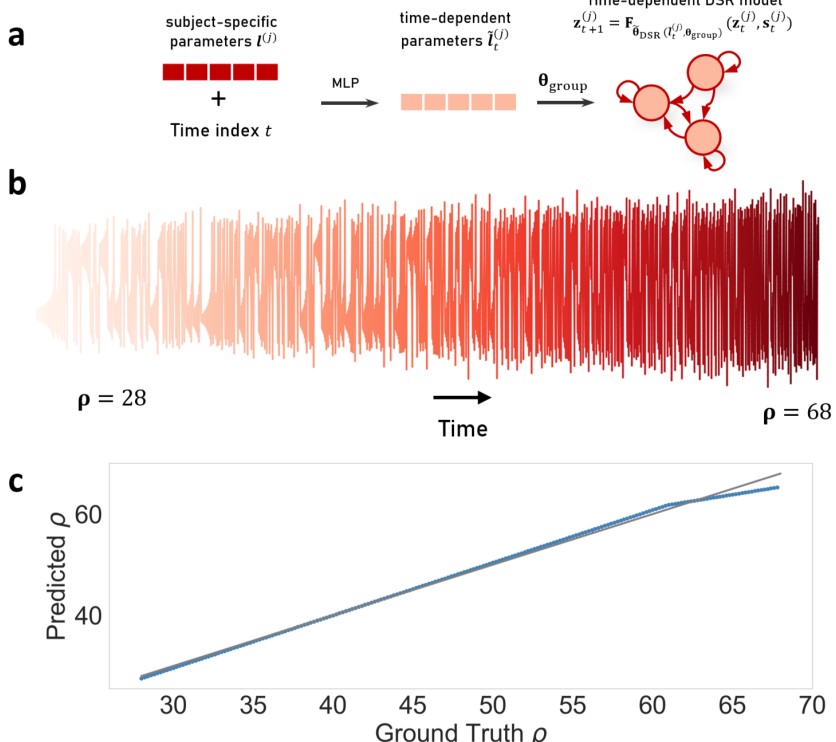

Figure 16: Augmenting the hierarchical shPLRNN for identifying drivers of non-autonomous dynamics: Making system-specific features time-dependent. (**a**) A multi-layer perceptron (MLP), mapping time and subject-specific feature vectors $l^{(j)}$ onto a time-dependent feature vector $\tilde{l}_t^{(j)}$, is jointly trained with the hier-shPLRNN, resulting in explicitly time-dependent parameters of the DSR model. (**b**) To test this idea, we simulated a simple setup where the control parameter $\rho$ of the Lorenz-63 system was linearly increased in the range $[28, 68]$, mimicking a time-dependent driving of the underlying system. For purposes of illustration, a hier-shPLRNN of just one subject was trained on time series segments from this simulation, but the approach could easily be extended to multiple subjects. (**c**) The temporal drift in $\rho$ was successfully recovered by the hier-shPLRNN, with a linear relation between actual and predicted $\rho$ ($R^2 = 0.996$). This simple example provides a proof of concept, but requires more detailed study to ensure this is a viable approach for more complex underlying systems and time dependencies in control parameters.

## A.3 DATASETS

**Lorenz-63 system** The famous 3d chaotic Lorenz attractor (Lorenz, 1963) is probably the most widely used benchmark system in DSR. Its governing equations are given by:

$$
\begin{aligned}
\dot{x} &= \sigma(y - x), \\
\dot{y} &= x(\rho - z) - y, \\
\dot{z} &= xy - \beta z.
\end{aligned}
\tag{15}
$$

Standard parameter settings putting the system into a chaotic regime are $\sigma = 10, \rho = 28$, and $\beta = 8/3$. We further included a Gaussian observation noise term, adding $5\%$ of the subject-specific data variance to each time series $x_{1\dots T_{\max}^{(j)}}^{(j)}$.

**Rössler system** The chaotic Rössler system (Rössler, 1976) is defined by:

$$\dot{x} = -y - z, \tag{16}$$
$$\dot{y} = x + ay,$$
$$\dot{z} = b + z(x - c).$$

Standard parameter settings that place the system in a chaotic regime are $a = 0.2$, $b = 0.2$, and $c = 5.7$. As for the Lorenz-63, we included a Gaussian observation noise term with $5\%$ of the subject-specific data variance.

**Lorenz-96 system**   The Lorenz-96 is an extension of the Loren-63 system to an arbitrary number of spatial dimensions $N$, introduced in Lorenz (1996). The governing equations are defined by

$$\dot{x}_i = (x_{i+1} - x_{i-2})x_{i-1} - x_i + F \tag{17}$$

with $x_{-1} = x_{N-1}$, $x_0 = x_N$, $x_{N+1} = x_1$, and a forcing parameter $F$. Again, we added Gaussian observation noise with $5\%$ of the subject-specific data variance.

Trajectories for all three benchmark systems were generated by numerically integrating the ODEs using the `odeint` function in `scipy.integrate` which implements the lsoda solver with a step size of $\Delta t = .01$.

**Double Scroll (Chua's Circuit)**   The double-scroll attractor, often studied with Chua's Circuit (Chua et al., 1986), is a 3-dimensional chaotic system. Its governing equations are given by:

$$\dot{x} = \alpha \left( y - x - f(x) \right), \tag{18}$$
$$\dot{y} = x - y + z,$$
$$\dot{z} = -\beta y,$$

where the piecewise-linear function $f(x)$ is defined as:

$$f(x) = m_1 x + \frac{1}{2}(m_0 - m_1)\left( |x + 1| - |x - 1| \right). \tag{19}$$

Standard parameter settings for a chaotic regime are $\alpha = 9.0$, $\beta = 14.0$, $m_0 = -1.143$, and $m_1 = -0.714$. For the results in Fig. 28, we chose $\alpha \in [8.4, 9.4]$, leading to both single-scroll and double-scroll behavior.

**Pulse Wave Dataset**   The pulse wave dataset (Charlton et al., 2019) was taken from `https://peterhcharlton.github.io/pwdb/`. It consists of pulse wave time series, simulated for 4,374 healthy adults aged 25–75 years, at 13 common arterial sites within the cardiovascular system (AorticRoot, ThorAorta, AbdAorta, IliacBif, Carotid, SupTemporal, SupMidCerebral, Brachial, Radial, Digital, CommonIliac, Femoral, AntTibial). For each site, four physiological quantities (Arterial Pressure Wave, Arterial Flow Velocity Wave, Arterial Luminal Area Wave, and Photoplethysmogram Wave) were simulated. This resulted in a total of 52 simulated pulse waves (4 quantities recorded at 13 sites) per virtual subject. The dataset further contains 32 hemodynamic parameters used for simulating the subject's pulse waves, listed in Table 6. The parameter PFT (time of peak flow at the aortic root) was nearly constant (78 for 124 subjects and 80 for all other subjects) and hence omitted from our regression analysis. For training, we standardized the time series subject- and dimension-wise, since some time series featured significant differences between subjects in their first moment, particularly in the Arterial Luminal Area Waves (possibly overriding some of the more relevant DS features as evidenced in lower training performance).

**Human EEG Data**   The EEG recordings from epileptic patients and healthy controls were originally provided in Andrzejak et al. (2001) and reformatted by Zhang et al. (2022). The dataset is publicly available on the Time Series Classification Website (`https://www.timeseriesclassification.com/description.php?Dataset=Epilepsy2`) under 'Epilepsy2'. The original set contains 500 single-channel EEG

| Abbreviation | Parameter |
|---|---|
| age | Age, in years |
| HR | Heart rate (HR), in beats per minute (bpm) |
| SV | Stroke volume (SV), in millilitres (ml) |
| CO | Cardiac output (CO), in litres per minute (L/min) |
| LVET | Left ventricular ejection time (LVET), in milliseconds (ms) |
| dp/dt | Max. value of first derivative of the aortic root pressure wave (mmHg/s) |
| PFT | Time of peak flow at aortic root, in milliseconds (ms) |
| RFV | Reverse flow volume at aortic root, in millilitres (ml) |
| SBP | Systolic blood pressure, in mmHg |
| DBP | Diastolic blood pressure, in mmHg |
| MBP | Mean blood pressure, in mmHg |
| PP | Pulse pressure, in mmHg |
| PP_amp | Pulse pressure amplification, a ratio of brachial to aortic PP |
| AP | Augmentation pressure at the carotid artery, in mmHg |
| AIx | Augmentation index at the carotid artery, % |
| Tr | Time to reflected wave at carotid artery, in ms |
| PWV_a | Pulse wave velocity (m/s) measured between the aortic root and iliac bifurcation |
| PWV_cf | Pulse wave velocity (m/s) measured between the carotid and femoral arteries |
| PWV_br | Pulse wave velocity (m/s) measured between the brachial and radial arteries |
| PWV_fa | Pulse wave velocity (m/s) measured between the femoral and anterior tibial (ankle) arteries |
| dia_asca | Ascending aorta diameter (mm) |
| dia_dta | Descending thoracic aorta diameter (mm) |
| dia_abda | Abdominal aorta diameter (mm) |
| dia_car | Carotid diameter (mm) |
| len | Length of proximal aorta (mm) |
| drop_fin | MBP drop from aortic root to digital artery, in mmHg |
| drop_ankle | MBP drop from aortic root to anterior tibial artery, in mmHg |
| SVR | Systemic vascular resistance ($\times 10^6$ Pa s / m$^3$) |

Table 6: The 32 ground truth haemodynamic parameters for the pulse wave dataset (Charlton et al., 2019). Descriptions taken from https://github.com/peterhcharlton/pwdb/wiki/pwdb_haemod_params.csv.

measurements, recorded for 23.6s at 174Hz. These were then split into 1s sequences, which were labeled as displaying either epileptic or healthy activity. From these sequences, 80 (40 per class) were selected as the training set on the Time Series Classification Website, which we used for our experiments. We further standardized the data with a global mean and standard deviation across all subjects in order to retain the relative amplitudes of the sequences.

**Human fMRI Data**    The fMRI dataset analyzed in this study was initially collected by Koppe et al. (2014), re-analyzed in Koppe et al. (2019), and made publicly accessible as part of the latter work. It comprises time series from 10 brain regions bilaterally, chosen for their consistent involvement in n-back working memory tasks as identified in a meta-analysis by Owen et al. (2005). Eigenvariates for these regions were extracted using the Brodmann area (BA) atlas. Specifically, the regions included medial and inferior parietal areas (BA7, BA40), prefrontal regions such as the dorsolateral prefrontal cortex (BA9, BA46) and ventrolateral prefrontal cortex (BA45, BA47), as well as the frontal pole (BA10), supplementary motor area (BA6), dorsal anterior cingulate cortex (BA32), and the cerebellum.

**Electricity Load Data**    The dataset from Trindade (2015) consists of electricity load time series from 370 sites recorded between 2011 and 2014. Each time series was preprocessed by removing leading zeros to account for differing start times across sites, resulting in varying maximum time lengths ($T_{\max(j)}$). The time series were standardized and slightly smoothed using a Gaussian kernel (window size = 5) and delay-embedded with a dimension of $d = 5$ and a lag of 10. We trained a feature vector with $N_{\text{feat}} = 12$ for every site.

## A.4    COMPARISON METHODS

**Selection of comparisons**    The approaches by Roeder et al. (2019), Yin et al. (2021) and Kirchmeyer et al. (2022) are the most sensible benchmarks for our study, as they address both forecasting and generalization across different parameter settings of DS. Yin et al. (2021) and Kirchmeyer et al. (2022) were directly evaluated in our work, while Roeder et al. (2019) was excluded due to their focus on short-term dynamical responses with relatively simple dynamics, and the integration of prior knowledge. Inubushi and Goto (2020) and Guo et al. (2021) do not learn subject-specific models or focus solely on shared dynamics, limiting their relevance for DSR. For Bird and Williams (2019), the primary focus was not on DSR but rather on generating sequences in specific styles using a multi-task DS. Further, no code was publicly available. Desai et al. (2022), on the other hand, requires strong

physical priors, making it less suitable for our benchmarks, which focus on more general, data-driven methods.

**LEADS** The LEADS (LEarning Across Dynamical Systems, (Yin et al., 2021)) framework decomposes the dynamics model into shared and environment-specific components. For each environment $e \in E$, the dynamics are modeled as a linear combination of shared dynamics $f$ and environment-specific dynamics $g_e$:

$$f_e(x) = f(x) + g_e(x), \tag{20}$$

where $f$ captures common dynamics across environments, and $g_e$ models the individual characteristics of environment $e$. The learning objective is to minimize the complexity of the environment-specific terms $g_e$ while ensuring accurate modeling of the overall system dynamics. This is achieved by solving the following optimization problem:

$$\min_{f, \{g_e\}_{e \in E}} \sum_{e \in E} \Omega(g_e) \quad \text{subject to} \quad \forall x_t^e, \frac{dx_t^e}{dt} = f(x_t^e) + g_e(x_t^e) \tag{21}$$

where $\Omega(g_e)$ is a regularization term that penalizes the complexity of the environment-specific terms.

We used the implementation of LEADS provided by the authors on their public GitHub page (https://github.com/yuan-yin/LEADS). For both the Lorenz-63 and Lorenz-96 benchmarks described in Sect. 4.1, we scanned different sequence lengths (seq_len $\in \{5, 10, 20, 30\}$, using a fixed time step dt $= 0.01$ and adjusting num_traj_per_env to provide the same number of training time steps per environment/setting as for the other methods. We used the Forecaster net with the 'mlp' setting and a hidden dimension of 6/20 (for Lorenz-63/96) to ensure approximately the same number of parameters as for the hier-shPLRNN. We used the default learning rate of $lr = 1.e-3$ as employed for all experiments in Yin et al. (2021), and scanned lambda_inv $\in \{5.e-4, 1.e-4, 1.e-5\}$ and factor_lip $\in \{1.e-2, 1.e-3, 1.e-4\}$. Despite hyperparameter tuning, LEADS struggled to capture the long-term dynamics of both the Lorenz-63 and Lorenz-96 systems. Models either converged to simple fixed points or exhibited unstable and divergent behavior for long-term forecasts. Given that performance metrics cannot be computed for divergent trajectories, the results presented here are hence for models predicting fixed point dynamics after short transients locally fitting the dynamics.

**CoDA** As in LEADS, in CoDA (Context-Informed Dynamics Adaptation, Kirchmeyer et al. (2022)) the dynamics for each environment $e \in E$ is modeled as a combination of shared dynamics $f$ and environment-specific dynamics. Specifically, a low-dimensional context vector $\boldsymbol{\xi}_e$ is used to generate parameter variations via a hypernetwork, resulting in environment-specific parameters:

$$\boldsymbol{\theta}_e = \boldsymbol{\theta}_c + \boldsymbol{W}\boldsymbol{\xi}_e \tag{22}$$

where $\boldsymbol{\theta}_c$ are the shared parameters, $\boldsymbol{W}$ is a learned weight matrix, and $\boldsymbol{\xi}_e$ is the environment-specific context vector. The dynamics for each environment $e$ are then given by:

$$f_e(x) = f(x; \boldsymbol{\theta}_e) = f(x; \boldsymbol{\theta}_c + \boldsymbol{W}\boldsymbol{\xi}_e). \tag{23}$$

As for LEADS, the learning objective is to minimize the contribution of the context vectors $\boldsymbol{\xi}_e$ while ensuring accurate modeling of the individual system dynamics. This is achieved by solving the following optimization problem:

$$\min_{f, \boldsymbol{W}, \{\boldsymbol{\xi}_e\}_{e \in E}} \sum_{e \in E} \left( L(f(x) + \boldsymbol{W}\boldsymbol{\xi}_e) + \lambda \|\boldsymbol{\xi}_e\|^2 \right), \tag{24}$$

where $\lambda \|\boldsymbol{\xi}_e\|^2$ is a regularization term. We used the implementation of CoDA provided on the authors' GitHub site (https://github.com/yuan-yin/CoDA). The architecture is fixed, allowing only to change the various regularization coefficients as hyperparameters. For those we scanned similar values as the authors in their experiments, $\lambda_{\boldsymbol{\xi}} \in \{1.e-2, 1.e-3, 1.e-4\}$, $\lambda_{\ell_1} \in \{1.e-5, 1.e-6, 1.e-7\}$ $\lambda_{\ell_2} \in \{1.e-5, 1.e-6, 1.e-7\}$, and $\dim(\boldsymbol{\xi}) \in \{3, 6, 10\}$. While overall working much better than LEADS, CoDA was still less reliable in recovering the long-term properties of the underlying systems than our method. Fig. 17 illustrates this for an example subject trained on the Lorenz-63, where the power spectrum learned by CoDA is much less accurate, especially in the low-frequency range.

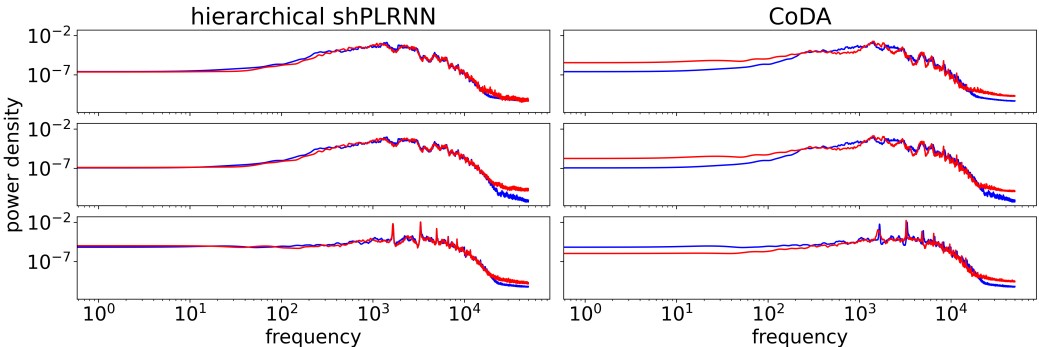

Figure 17: Power spectra of ground truth (blue) and generated (red) trajectories of one of the Lorenz-63 settings for a hierarchical shPLRNN and CoDA.

## A.5 UNSUPERVISED FEATURE EXTRACTION

**tsfresh**  We used the Python implementation of `tsfresh` (Christ et al., 2018), which automatically extracts a large set of time series features and selects the most relevant ones for further analysis.

**Catch-22**  Catch-22 is a collection of 22 time series features that are both informative and computationally efficient (Lubba et al., 2019). We used the default implementation from the `tslearn` package.

**ROCKET and MiniRocket**  ROCKET (RandOm Convolutional KErnel Transform), and its variant MiniRocket, use random convolutional kernels to project time series into high-dimensional feature spaces useful for linear classification and unsupervised feature extraction (Dempster et al., 2020; 2021). Here we used the default setting of $10,000$ random kernels in the `tslearn` package, leading to a $20,000$-dimensional embedding space.

**Attention-based Convolutional Autoencoders**  Lastly, we trained an attention-based convolutional autoencoder (Wang et al., 2023), alternating convolutional layers with time and feature attention layers in sequential order. We fitted a GMM for unsupervised cluster assignment to the latent representations learned by the encoder, where we chose the size of the encoding space to be the same as that of the hier-shPLRNN feature space ($N_{\text{feat}} = 10$).

Table 7: Average classification accuracy of Gaussian Mixture Models (GMMs) applied to features extracted through various unsupervised methods on the EEG data. Values are across 10 training runs for each model. A randomly initialized GMM was fitted for each training run, so variation in deterministic models stems from the different GMM fittings, while in 'non-deterministic models' (hier-shPLRNN, MiniRocket, Attn-Conv-AE) it comes from both training algorithm and GMM fitting.

| Model | Classification Accuracy |
|---|---|
| hier-shPLRNN | **$92.6 \pm 1.2\%$** |
| ROCKET (Dempster et al., 2020) | $71.25 \pm 2.2\%$ |
| MiniRocket (Dempster et al., 2021) | $65 \pm 13\%$ |
| tsfresh (Christ et al., 2018) | $68.1 \pm 3.3\%$ |
| Catch-22 (Lubba et al., 2019) | $57.6 \pm 2.5\%$ |
| Attn-Conv-AE (Wang et al., 2023) | $71.5 \pm 1.8\%$ |

### A.5.1 HIERARCHICAL MARKOV NEURAL OPERATOR

To illustrate the wider applicability of our approach, we also tested the hierarchization framework with the Markov Neural Operator (MNO) suggested in Li et al. (2022). Code is publically available at https://github.com/neuraloperator/markov_neural_operator. As described in Sect. 3, we employed low-dimensional feature vectors and higher-dimensional group-level matrices linearly mapping onto all the trainable parameters of the MNO. We trained with $N_{feat} = 1$ on the

Lorenz-63 setup from sect. 4.2, and batch size, learning rates and initialization chosen as discussed in Appx. A.1. Following Li et al. (2022), we trained on one-step-ahead predictions, combined with a physics-informed dissipative loss. This loss enforces dissipativity by penalizing deviations from the global attractor through a regularization term that mimics energy dissipation. This initial implementation and the results in Fig. 18 are meant to highlight the straightforward applicability of the hierarchical framework to other architectures.

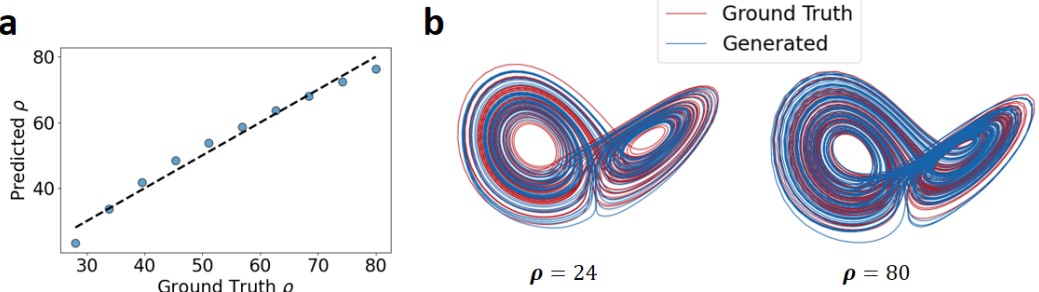

Figure 18: **a**: Ground truth and predicted $\rho$ from $1d$ feature values for the hierarchical MNO model (linear fit: $R^2 = 0.97$), similar to the results in Figs. 3 & 5. **b**: Long term freely generated dynamics for different $\rho$ values.

## B  FURTHER RESULTS

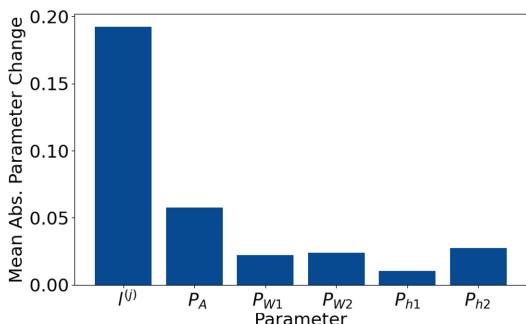

Figure 19: Average parameter changes during training of a hierarchical model on time series of the Lorenz-63 system. Subject-specific feature vectors ($l^{(j)}$) change much more than group-level parameters ($\boldsymbol{P}_A, \boldsymbol{P}_{W_1}, \boldsymbol{P}_{W_2}, \boldsymbol{P}_{h_1}, \boldsymbol{P}_{h_2}$).

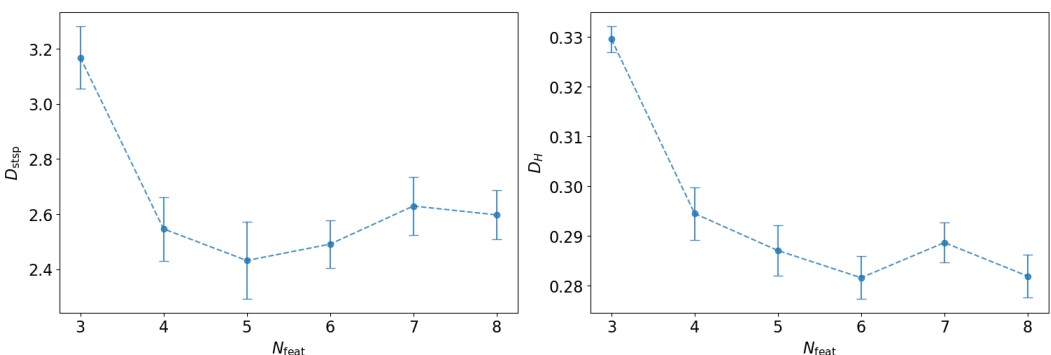

Figure 20: Reconstruction performance as a function of $N_{\text{feat}}$ for models trained on the 64-subject Lorenz-63 dataset. The feature vector must be large enough to encode the variation in subject-specific dynamics when changing all three of the Lorenz-63 bifurcation parameters (but may not be too large either).

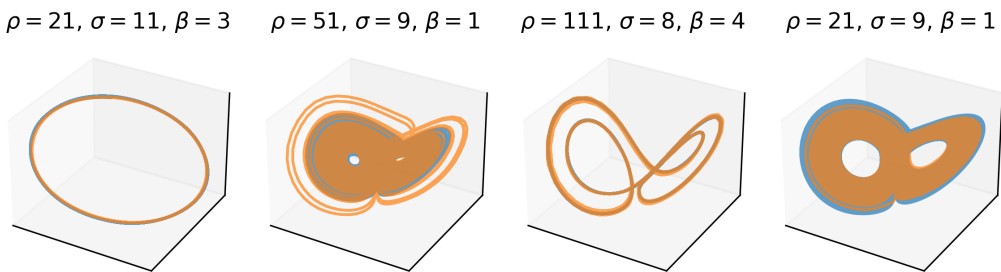

Figure 21: Ground truth (blue) and hier-shPLRNN generated (orange) trajectories for the Lorenz-63 system. Shown are four example parameter combinations, illustrating different dynamical regimes of the system. All of these were generated by the same hier-shPLRNN model.

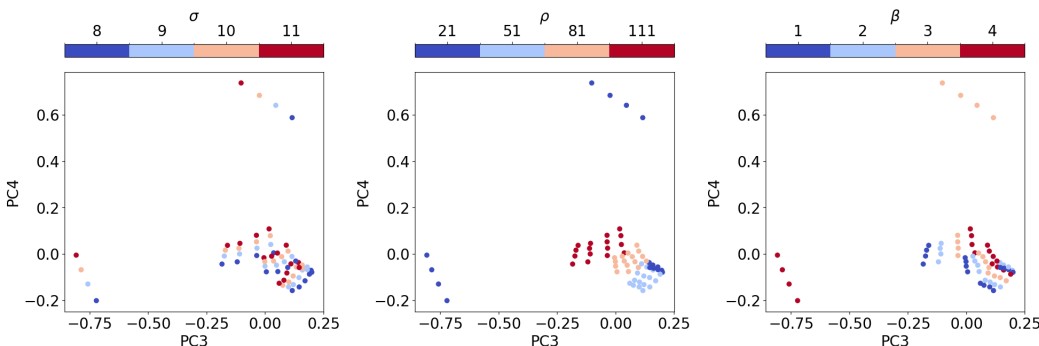

Figure 22: Same as Fig. 4 for the third and fourth principle component.

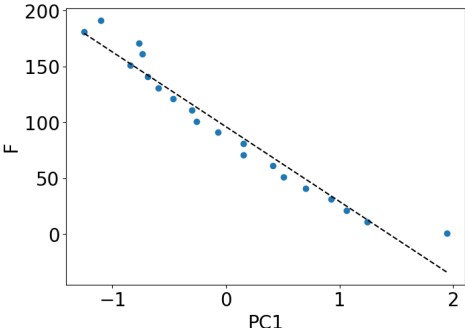

Figure 23: First PC of the 5d feature vector of a hier-shPLRNN, trained on the Lorenz-96 system in Sect. 4.1 aligns with the ground truth forcing parameter $F$ used to generate the training data.

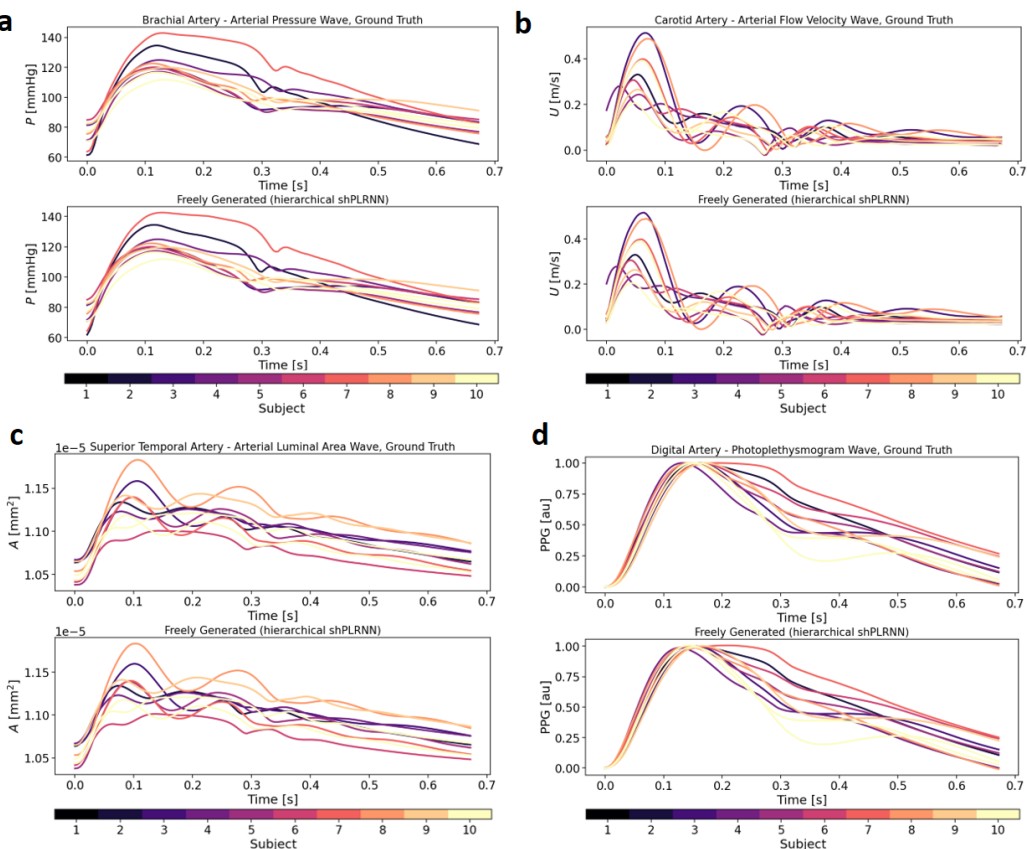

Figure 24: Ground truth (top) and freely generated (bottom) arterial pressure wave (**a**), arterial flow velocity wave (**b**), arterial luminal area wave (**c**) and photoplethysmogram wave (**d**), simulated at 4 different anatomical sites, for 10 representative subjects (selected by k-medoids, color-coded).

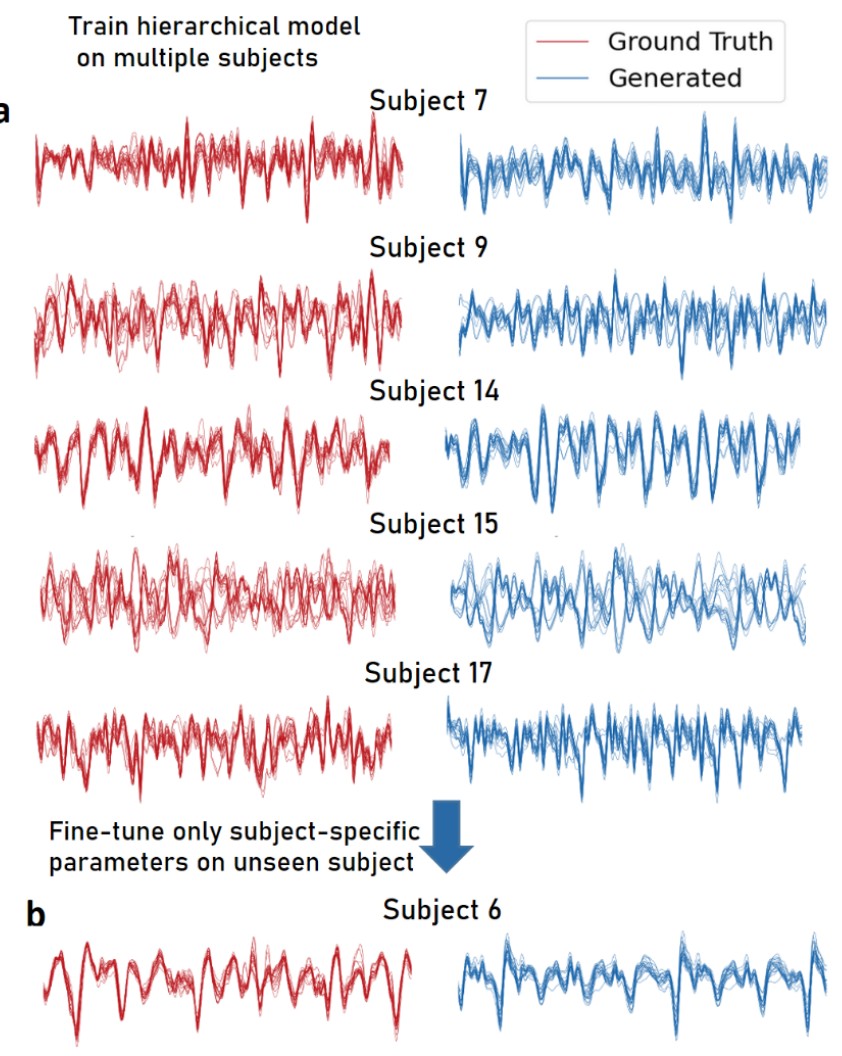

Figure 25: Example reconstructions of several subjects from the experimental fMRI dataset using a hier-shPLRNN with $N_{\text{feat}} = 10$, $L = 300$, and training by GTF with $\alpha = 0.1$.

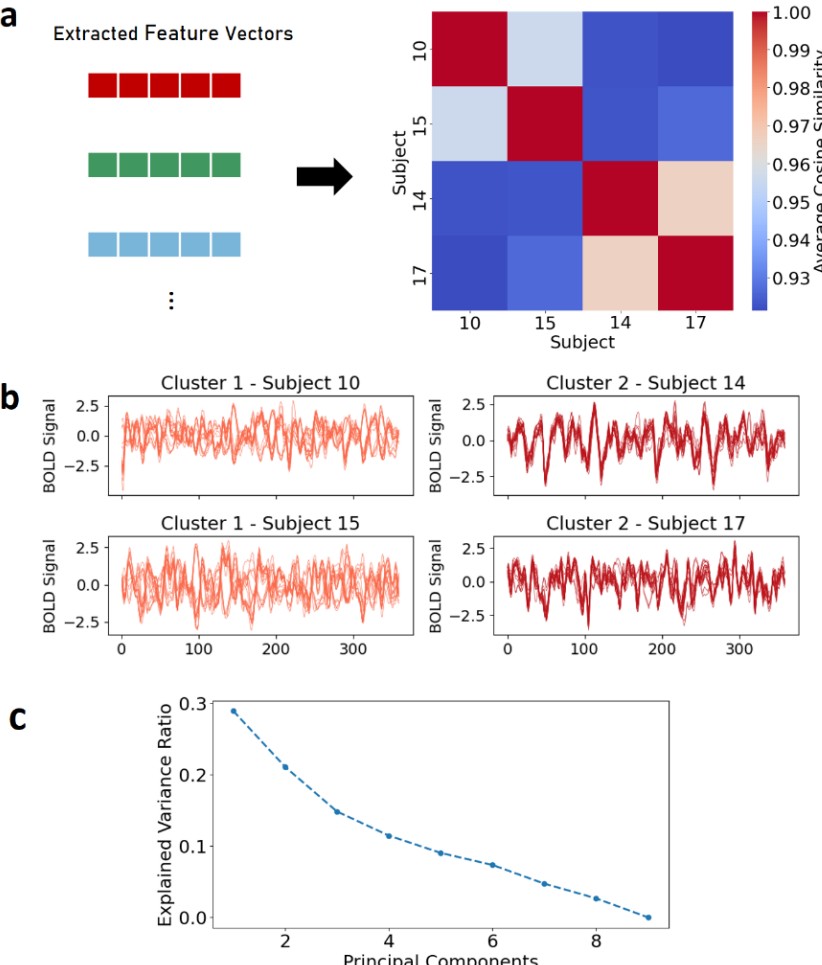

Figure 26: **a**: Cosine similarity matrix based on the average similarity of the feature vectors across 20 training runs for 4 example subjects. **b**: Extracted similarities and cluster labels reflect visual differences in the recorded BOLD signals. Cluster assignments were consistent across all 20 runs for all subjects except one (subject 7), for whom the cluster assignment alternated between both clusters with a 50/50 split. **c**: Explained variance ratio for the principal components obtained from a PCA on the subject feature space for a model trained on the 10 fMRI subjects.

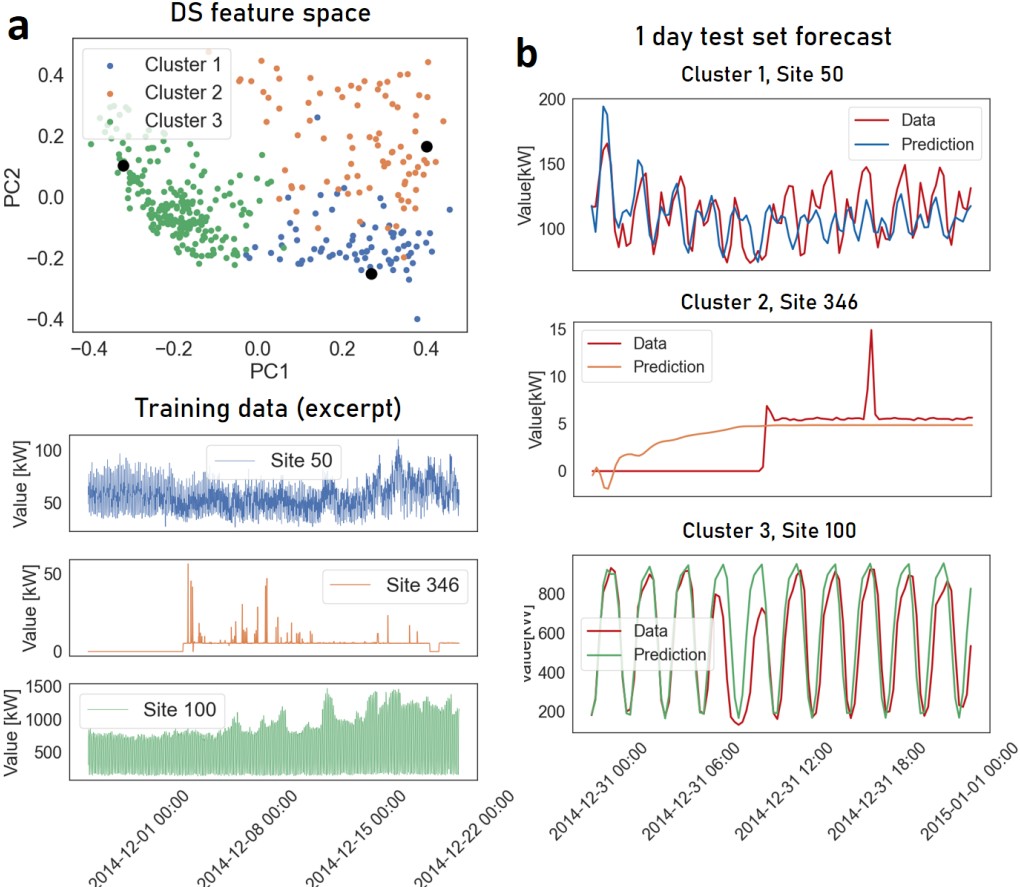

Figure 27: **a**: Visualization of the first two PCs of the $12d$ inferred DS feature space for the electricity load data. Points represent sites, color-coded by cluster. Highlighted sites (black dots) correspond to examples shown below. Training data excerpts from three example sites illustrate pronounced differences in time series and highlight the highly nonstationary nature of the data. **b**: One-day test set forecast for each representative site, comparing ground truth (red) and model predictions (blue/orange/green). Predictions sometimes closely match ground truth values while capturing site-specific variations in complexity.

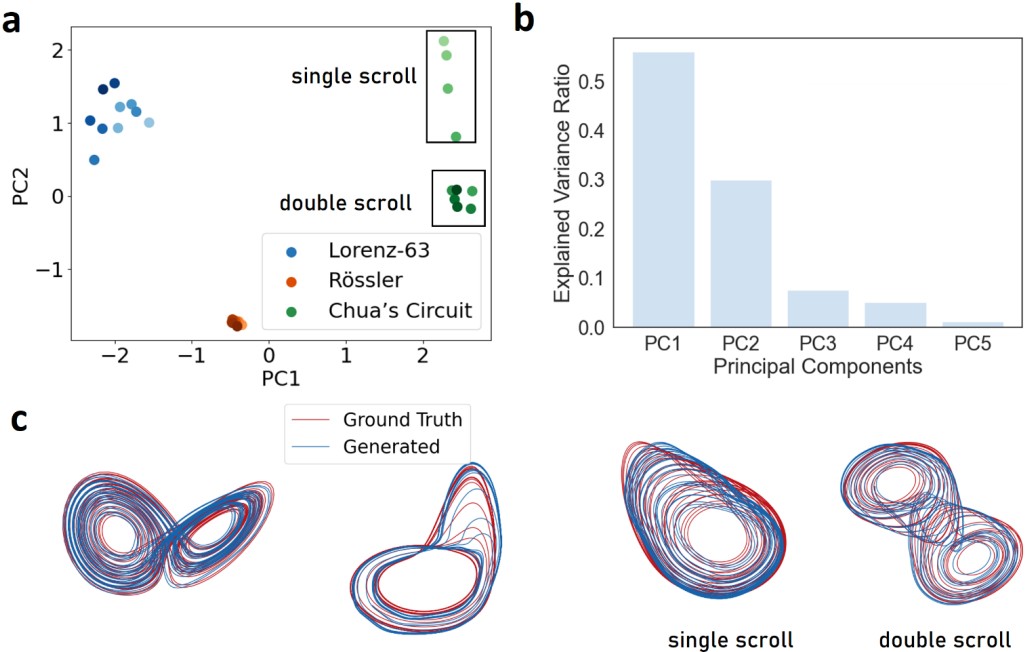

Figure 28: **a**: First 2 principal components (PCs) of the DS feature space for a model jointly trained on observations from the Lorenz-63 system, Rössler system, and Chua circuit, with 10 different values each for the GT parameters $\rho$, $c$, and $\alpha$. The different DS are clearly separated along PC1, while PC2 distinguishes between the single-scroll versus double-scroll behavior of the Chua circuit (see **c**). **b**: Despite joint training on three different DS, the hier-shPLRNN embeds features in a low-dimensional space where the first four PCs capture almost all variation across subjects. **c**: Example reconstructions for all three systems.

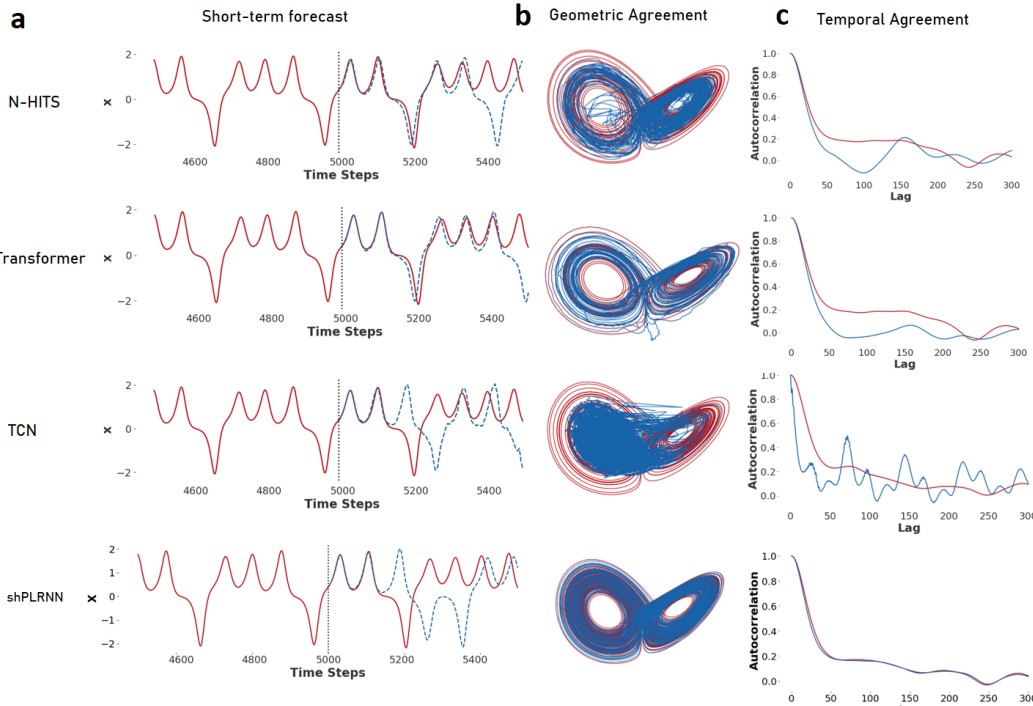

Figure 29: Comparison of several popular time series (TS) models, N-HITS (Challu et al., 2022), Transformers (Vaswani et al., 2017), and Temporal Convolutional Networks (TCNs) (Bai et al., 2018), to the shPLRNN trained with GTF (Hess et al., 2023) with respect to forecasting performance (**a**) and long-term statistics (**b**, **c**) on observations ($T_{max} = 5000$) from the chaotic Lorenz-63 attractor (Eq. 15). TS models were trained for 500 epochs using their implementations in the `darts` package (Herzen et al., 2022), with an input window size of 100 time steps, to forecast 30 time steps. All TS models had around $40,000$ parameters, while the shPLRNN only required 706 trainable parameters in the chosen configuration ($M = 3, L = 100$). Long-term forecasts were generated – similar to the shPLRNN – by recursively feeding the TS models their own predictions. **a**: Short-term test set forecasts generated by the TSF models often match (or even exceed) the accuracy of the shPLRNN. The somewhat better forecasting performance is easily explained by the longer initialization windows used for the TS models compared to the shPLRNN (and potentially the much larger number of trainable parameters), which is initialized using just a single time step $x_0$ at the beginning of the test set (in line with a state space's Markov property in DS theory). **b**: However, unlike the shPLRNN, TS models fail to reconstruct the Lorenz-63 attractor geometry. **c**: Likewise, and unlike the shPLRNN, their long-term forecasts completely fail to capture the statistical temporal structure of the data, assessed here via the autocorrelation function (Wood, 2010) over the first 300 time lags.

