# OpenReview forum: "Learning Interpretable Hierarchical Dynamical Systems Models from Time Series Data"
_ICLR.cc/2025/Conference — ICLR 2025 Poster_

### Official Review · Reviewer_Qfhy · 2024-10-19

**Soundness:** 4
**Presentation:** 3
**Contribution:** 4
**Rating:** 8
**Confidence:** 4

**Summary:**

The authors introduce a general theoretical problem: reconstructing a set of dynamical systems, given short observations of each system with different underlying parameters. The authors map this problem onto problems frequently arising in biophysics and neuroscience, in which one needs to train a joint model with subject-specific attributes. The authors introduce a hierarchization scheme for training recurrent neural networks, that disentangles low-dimensional subject-specific features from the general features of a shared dynamical systems model. The authors apply their approach first to synthetic dynamical systems tasks, then to an arterial pulse dataset, and finally to EEG and fMRI datasets. Beyond strongly outperforming baselines and ablations, the authors show that their approach extracts predictive dynamical features that can be used for interpretability, or even for subject classification.

**Strengths:**

This paper is very clear, thorough, and well-motivated. I think this paper clearly exceeds the acceptance bar for the conference.

+ The Introduction and Related Work sections are comprehensive and concise. The authors cover standard references for SciML (Koopman/PINN/neural ODE/RC), but they also highlight several hihgly-relevant but lesser-known methods, including recent works that focus specifically on reconstructing invariant measures of dynamical systems.

+ The baseline models chosen are reasonable and appropriate points of comparison. The authors include an explicit ablation of the hierarchical portion of their model

+ The two application tasks (EEG and fMRI) are important and provide clear motivation for this approach, which has an inductive bias for datasets with discrete subjects. Framing subject-level clinical data as analogous to training a joint model on copies of a dynamical system with different parameters is a very conceptually-rich approach.

+ There are lots of novel details, which show that the authors have thought very deeply about their applications. For example, the covariance-scaled MSE (Eq. 5) helps account for clinical batch effects.

+ The arterial pulse dataset is an excellent demonstration, showing that the model successfully finds a low-dimensional description of an otherwise complex and high-dimensional simulation. The regression showing that the fitted model predicts the haemodynamic parameters is a great demonstration of the interpretability of the learned features.

**Weaknesses:**

I’ve assigned a high score because I think this paper clearly meets standards of novelty, rigor, and interest for the conference. However, I have a few general concerns, which the authors might consider in order to attract a wider audience to their work:

+ The description of the training method itself could be expanded. I understand the architecture, and the use of a low-dimensional subject vector that linearly decodes into the high-dimensional weight vectors of the RNN. However, I do not understand the mechanics of how the subject-specific vectors are found and P matrices are trained. I understand that once we have the W matrices, then we can use generalized teacher forcing; it’s the step before that where I am unclear. I see this hierarchical training scheme as one of the main contributions of the paper, and so clarifying this is essential.

+ The shPLRN model is well-motivated for this problem and has been characterized extensively in prior work. However, I do perceive it as a bit of a specialized model, compared to more common approaches to DSR like PINNs or even SINDY. An example of the hierarchization scheme, which is general, used on one of these more common DSR approaches would be an interesting baseline (LEADS, a baseline currently included in the manuscript, is essentially this for neural ODEs).

My concern is that I perceive the primary contribution of this paper to be getting the hierarchization scheme, including the training approach (which may be hard to adapt to models that can’t be trained with teacher forcing). The other novelty is the choice of datasets and demonstrations, which are all interesting and strong. However, the current paper mixes these contribution with a specialized architecture from prior work, shPLRNN, making it hard to disentangle the contributions. My concern is that readers who encounter this paper will incorrectly think that the architecture itself is the main contribution, rather than the problem setup and training scheme.

**Questions:**

+ Eq. 5 — can you please clarify how this is used in the training loop? How and when is \Sigma updated? Generally I would prefer some more detail here.

+ Can you provide more description of how the Hierarchical training scheme works?

---

> ### Author Response · Authors · 2024-11-21
> **point-by-point reply**
>
> We thank the referee for the enthusiastic support of our work and the thoughtful comments!
>
> **Weaknesses**
>
> *W1 (description of training method)*: Thank you for raising this point, this was indeed described a bit too briefly. We have now thoroughly rewritten sect. 3 and expanded on these points; at the same time streamlined this sect. and moved details which are less essential (and may have disrupted the flow) to the Appx.. We also included a new figure (Fig. 8) to further clarify this aspect, and amended the model overview in Fig.1.
>
> In essence, the trainable parameters of the overall framework consist of (a) the group-level matrices $\\mathbf{P}\_{\\mathbf{A}}$, $\\mathbf{P}\_{\\mathbf{W}_1}$ etc. (as listed on lines 170-174), (b) the subject-specific features $\\mathbf{l}\_j, j=1...S$, and (c) scaling parameters $\\Sigma\^{(j)}$ in eq. 5. These are all trained simultaneously, end-to-end, on the MSE loss eq. 5. While training, GTF replaces subject-specific latent states in eq. 4 by an optimal linear mixture of model-forward-iterated states $\\mathbf{z}\_{t-1}$ and data inferred states $\\mathbf{z}\_{t-1}=\\mathbf{x}\_{t-1}$ as described in A.1.1. As each of the matrices in eq. 4 is a linear combination of group-level matrices $\\mathbf{P}$ and subject-specific features $\\mathbf{l}_j$ (i.e., could be rewritten in terms of those as on lines 172-174), qua chain rule GTF’s gradient control transfers directly to these matrices (see first pg. of sect. 3.2 and new Fig. 8).
>
> *W2 (alternative models)*: We had initially chosen the shPLRNN as our base model based on its SOTA performance compared to many other architectures, including SINDy (Hess et al. [1]). However, we completely acknowledge this point and addressed it by repeating the experiments in sect. 4.1 with vanilla RNNs (still trained by GTF) and LSTMs (as e.g. used in Vlachas et al. [2]) as DSR models, see Appx. A.1.6. New Table 3 confirms that the basic pattern of results stays the same.
>
> W.r.t. PINNs and SINDy in particular, however, note these are different from our approach (and from neural ODEs) in that they both require physical priors about the functional form of the underlying equations, which for complex systems like the EEG or fMRI data considered here is a bit difficult to specify. In fact, it has been shown previously that SINDy fails when its function library doesn’t match the ground truth (Göring et al. [4]) or in empirical scenarios where such priors are difficult to obtain (Hess et al. [1]; with related problems reported for PINNs in e.g. Krishnapriyan et al. [5]; we further observed that SINDy also runs into severe numerical issues on higher-dimensional datasets like those used here, with models practically always diverging).
>
> In general, a physical prior should also be sufficiently broad to accommodate *all* systems trained on, not just one particular system, a point on which we added a note to the Limitations sect.. One such more generic prior could be strong dissipativity in a class of DS. This assumption is used in Markov Neural Operators [4], and so we tested MNOs as an architecture with a physics-inspired regularization loss within our hierarchisation framework (see new Appx. A.5.1); new Fig. 17 essentially reproduces the observations from Fig. 3 with the MNO, likewise obtaining an about linear relation between the inferred feature value and the true control parameter.
>
> **Questions**
>
> *Q1 (Eq. 5)*: See our response to W1 above: The $\Sigma^{(j)}$ are trained simultaneously with all other model parameters, as we now clarified in the first pg. of revised sect. 3.2.
>
> *Q2 (how training works)*: We hope this is now sufficiently addressed by our response to W1 above, by the textual changes and additions we have done in this context (rewritten sect. 3.2), and the updates to Fig. 1 and the new Fig. 8.
>
> **References**
>
> [1] F. Hess et al. “Generalized Teacher Forcing for Learning Chaotic Dynamics,” in Proceedings of the 40th International Conference on Machine Learning
>
> [2] Vlachas et al., “Data-driven forecasting of high-dimensional chaotic systems with long short-term memory networks,” Proceedings of the Royal Society A: Mathematical, Physical and Engineering Sciences, vol. 474, no. 2213, p. 20170844, May 2018
>
> [3]  Brenner et al., “Almost-Linear RNNs Yield Highly Interpretable Symbolic Codes in Dynamical Systems Reconstruction,” Oct. 18, 2024, arXiv: arXiv:2410.14240
>
> [4] N. A. Göring et al.  “Out-of-Domain Generalization in Dynamical Systems Reconstruction,” in Proceedings of the 41st International Conference on Machine Learning
>
> [5] A. S. Krishnapriyan et al., “Characterizing possible failure modes in physics-informed neural networks,” in Advances in Neural Information Processing Systems, vol. 34, 2021.

---

> > ### Comment · Reviewer_Qfhy · 2024-11-25
> > **Thanks for detailed reply, my questions have all been addressed**
> >
> > Thanks very much for your detailed reply and revisions. I agree regarding the strong prior of SINDy/PINN approaches compared to neural ODE and RNN. All of my questions have been addressed.

---

> > > ### Author Response · Authors · 2024-11-30
> > >
> > > Thank you very much for taking the time to go through the thread with ref. D8qe, and for your support on this, much appreciated.

---

### Official Review · Reviewer_D8qe · 2024-11-01

**Soundness:** 2
**Presentation:** 2
**Contribution:** 2
**Rating:** 1
**Confidence:** 4

**Summary:**

This paper presents a dynamical modeling approach for time series generative foundation models, focusing on scientific applications. The authors argue that a robust time series generative model requires both in-domain and cross-domain features, achieved through hierarchical dynamical systems. They frame the construction of such models as a dynamical system reconstruction (DSR) problem, training RNNs with low-dimensional task-specific and high-dimensional group-specific features. The method is supported by numerical experiments.

**Strengths:**

Overall, this paper might present a good idea, but it seems rushed, which affects the contribution of an otherwise very interesting work. Regardless of the final decision, I hope the authors find my comments helpful in further refining their draft.

* Important modeling problem for scientific AI/ML/Foundation Models

* The experiments are solid

* Many ablation studies

**Weaknesses:**

* **Clarity.** The language is overly complex and difficult to read. For example, in lines `134-145`, the authors use 10 lines for just 3 sentences, making the paragraph hard to follow. I had to read Sec 3.1 multiple times and still found it difficult to understand.

* **Novelty.** The proposal lacks clear motivation and feels too straightforward to be considered novel. Most components are derived from existing work (e.g., General Teacher Forcing, PLRNN, shPLRNN).

* **Practicality.** The model's parameterization is overly complex (e.g., passing learnable parameters to the next layer), raising concerns about its feasibility for end-to-end training and practical use as a time series generative model.

* **Significance.** The significance is difficult to gauge due to limited clarity. Overall, it seems lacking, given the assembly-like nature of the approach.

---


### Update 11/30/2024: Changed score to strong reject due to unprofessional rebuttal.

**Questions:**

* Why Eqn 3 is mathematically tractable?

* Why does this tractability matter? It seems only loosely connected to the task of interest (e.g., certain physical systems' ODEs).

* Could you test your proposal on common time series tasks? For example, multivariate time series datasets commonly used in forecasting literature, such as traffic, electricity, exchange rates, etc. If not, could you provide some justification? As mentioned in your conclusion:
  > Unlike TS models, where the interest lies mainly in forecasting or TS classification/regression, in DSR we aim for generative models that capture invariants of the true

  Why wouldn’t these common forecasting tasks contain such "invariants of the truth"?

* This paper reminds me of an ICML 2023 paper: https://arxiv.org/abs/2306.06252. While the scope is different (they focus on feature extraction of time series by assuming the underlying dynamical system is a dynamical Ising model), they also attempt to interpret general time series through a dynamical systems lens. While the exact relevance is unclear, treating TS modeling as a dynamical system is not new. I am sure there are other works in this direction. It would be valuable to discuss more related literature in the ML community.

---

> ### Author Response · Authors · 2024-11-21
> **point-by-point reply, Weaknesses**
>
> We thank the referee for the general appreciation of our work, and the constructive comments.
>
> **Weaknesses**
>
> *W1 (clarity)*: We reworked major parts of the text, in particular completely rewrote the Methods sect. 3.1 & 3.2, in an attempt to streamline it and improve clarity (reducing sentence length, rearranging material, adding explanation, revising Fig. 1 & adding new Fig. 8; please compare revised to previous version). This is important, as we feel there were indeed some major misunderstandings, as explained in more detail in our responses below.
>
> *W2 (novelty)*: Our model is not primarily a time series (TS) model, but a dynamical systems reconstruction (DSR) approach. It is important to note that, while TS modeling and DSR are certainly related, they are not the same and have partly different objectives. DSR is by now a burgeoning field in its own right in scientific ML (as reviewed e.g. in [1,2]), a trend acknowledged by ICLR by including ‘dynamical systems’ as a topic term fairly recently. Unlike conventional TS modeling, in DSR the central goal is to learn from data a generative model of the *underlying system dynamics*, which means the model must also express similar *long-term statistics* (as assessed by, e.g., agreement in auto-corr functions or power spectra) and a similar state space geometry with the same attractor states (as assessed via our geometrical criterion $D_{stsp}$). This is crucially important for scientific purposes, where we seek a model that reflects the ground truth in its *dynamical mechanisms*. But it might be less relevant for conventional TS analysis, where we may be content with a model that produces good ahead predictions or downstream classification results.
>
> This crucial difference we have now illustrated in new Fig. 28, where we compared the shPLRNN trained with GTF to various popular TS models, including N-HITS, temporal CNNs, and Transformers. While all models produce reasonable short-term forecasts (up to the prediction horizon given by the exponential divergence of trajectories in chaotic systems), *only the shPLRNN trained by GTF* produces sensible reconstructions of attractor geometry and long-term statistics.
>
> To the best of our knowledge, ours is the first comprehensive hierarchical framework for DSR that allows for integration from many different systems, living in different dynamical regimes (e.g. with chaotic vs. periodic dynamics), into the same underlying latent model. The only somewhat related frameworks (LEADS and Coda) were profoundly out-performed by our construction, and often were not able to actually reconstruct the systems in question (see Table 1 \& Fig. 16). More importantly, we report an important connection between subject/system-specific feature vectors and crucial control parameters of the ground truth (GT) system. This semantic structuring of the feature space with respect to GT parameters that control transitions between different dynamical regimes (now backed up by further results on simultaneous observations from completely different systems, new Fig. 27), is, we think, extremely valuable for scientific/ medical interpretation. And, although these scientific applications were our prime objective, we believe it may also have strong implications for downstream classification tasks, usually a domain of classical TS models, as shown in Fig. 7.
>
> *W3 (practicality)*: We would argue that our model is far less complex, as, for instance, the Transformer architecture (Vaswani et al., [3]) with its many different mechanisms and components (temporal encoding, embeddings, encoder/ decoder/ MLP layers, self-/cross-attention etc.). In essence, there are parameters at only *two levels*, the group-level (common to all subjects) and the subject/system level (the subject-specific feature vectors). These are related to the NN parameters that instantiate the subject-level models (which all share the same architecture, eq. 4), through a simple matrix multiplication and thus just one step of the chain rule, computed by autograd [4]. Hopefully some of these aspects are clearer now after our comprehensive rewrite of sect. 3.
>
> The framework’s feasibility for end-to-end training on DSR tasks we believe we have thoroughly demonstrated in Figs. 2-7, 23-26, using a variety of different benchmarks and challenging real-world datasets (see also Table 1).

---

> > ### Author Response · Authors · 2024-11-21
> > **Weaknesses (continued) & Questions**
> >
> > *W4 (significance)*: See our reply to W2 above. We believe the perceived lack of significance is related to the misinterpretation of our framework as one for TS modeling (which may have been our fault in not conveying the goals of our work clearly enough). This essentially ignores that our framework delivers subject-level reconstructions of the full system dynamics, which, beyond pure TS modeling, agree with the GT in their dynamical mechanisms, attractor geometries, and *long-term* behavior (cf. Fig. 28). *Only through this*, the subject-level features obtain their interpretation in terms of underlying *control parameters*, a result that could not have been achieved by mere TS modeling (because it does not reconstruct the dynamics, Fig. 28).
> >
> > **Questions**
> >
> > *Q1 (tractability of Eq. 3)*: Eq. 3 is tractable in a *dynamical system’s sense* (see our replies above): Its piecewise linear nature (simply a consequence of the ReLU) enables to compute important dynamical objects like fixed points or cycles exactly and semi-analytically, and thus to analyze topological properties of the model’s state spaces more rigorously; see [5] for an extensive treatment of how to compute topological properties and bifurcation curves for these types of models.
> >
> > *Q2 (importance of tractability)*: Tractability is essential for scientific insight, where beyond just prediction or classification (as in conventional TS modeling), we are interested in *dynamical mechanisms* underlying the observed time series. The model’s tractability enables to efficiently compute attractors and other invariant sets in the system’s state space [1,5], which explain the system’s long-term behavior, temporal characteristics, or its sensitivity to perturbations, for instance. Such analyses are frequently performed in theoretical neuroscience, for example, to identify computations underlying behavior (e.g. [22], [23]), but also in ML (e.g. [24]). Piecewise linear models therefore are very popular in the scientific ML and DSR literature (e.g. [6-11]).
> >
> > *Q3 (common TS tasks)*: For the reasons noted above our focus was on popular benchmark tasks in the *DSR literature* (like the Lorenz-63 system) and on *scientific* applications (like the fMRI/ EEG data). With ‘invariants’ we meant invariant statistical and geometrical properties of the dynamics, like attractor states (formally sets $S$ in state space that are invariant w.r.t. the system’s flow operator $\Phi$, i.e. $\Phi(S) \subseteq S$, see e.g. textbooks by Perko  [12] or Guckenheimer & Holmes  [13]). This is commonly not assessed in conventional TS models, and a number of recent publications ([15-19], overview in [1]) have demonstrated that conventional training techniques like BPTT are not sufficient for capturing such invariant properties, but either these need to be built into the loss function [15-18] or specific control-theoretic training techniques [11,19] are required.
> >
> > As noted in our reply to W2 above, we have now illustrated this important point in Fig. 28, where we compared several popular TS models (N-HITS [20], Transformers [4] and Temporal Convolutional Networks [21]) to the shPLRNN trained by GTF in terms of both ‘traditional’ forecasting and of reconstruction of long-term statistics and geometrical properties. Fig. 7 further supports the idea that some downstream tasks like unsupervised clustering may become a lot easier in a space spanned by dynamical systems rather than more conventional TS features.
> >
> > We also took up the referee’s suggestion and tested our framework now on a dataset perhaps more popular in the TS modeling literature, namely time series for electricity consumption at 370 different sites. New Fig. 26 shows that our model also extracts a meaningful feature space from this dataset, and (despite not being explicitly built for forecasting, but for DSR) achieves good predictions on unseen test data.

---

> > > ### Author Response · Authors · 2024-11-21
> > > **Questions (continued) & References**
> > >
> > > *Q4 (related lit.)*: Yes, indeed, there is of course related literature, but we believe we have covered most of it in our Related Works sect. 2 (which referees LBGT and Qfhy explicitly acknowledged as being comprehensive and extensive)? Nevertheless, we have updated this sect. with further references. We have also benchmarked against many of these related approaches, see for instance Table 1 and Table 3. But coming back to the points we made above in response to W2 & Q3, please note that DSR goes beyond TS modeling and just interpreting time series as coming from a dynamical system, or adopting such a perspective (this view is almost as old as dynamical systems theory itself). DSR is much more explicitly concerned with *recovering attractor geometries and the system’s long-term statistical (ergodic) properties from the data* (Fig. 28), aims which only have been achieved in recent years through novel training and modeling approaches.
> > >
> > > The study cited by the referee, for instance, although involving a dynamical model, is not a DSR approach (and thus, we feel, not strongly related to our work). The Ising model also comes with a potential function (or in this probabilistic case with an equilibrium, Boltzmann, distribution), and as such, similar to a Hopfield network, cannot express many dynamical phenomena of interest, incl. essentially all the chaotic dynamical systems considered in our paper (see, e.g., textbooks by Perko [12], Strogatz [14], on systems with a potential/ Hamiltonian).
> > >
> > > We hope these points are clearer now in our revision, where we tried to emphasize more the aims of the field, within which our approach is situated, and added an illustrative example in Fig. 28 of one of the crucial differences between TS modeling and DSR (another one is in fact model tractability/ interpretability, which plays a bigger role in DSR with scientific applications in mind).
> > >
> > > **References**
> > >
> > > [1] D. Durstewitz et al., “Reconstructing computational system dynamics from neural data with recurrent neural networks,” Nat. Rev. Neurosci., vol. 24, no. 11, pp. 693–710, Nov. 2023, doi: 10.1038/s41583-023-00740-7.
> > >
> > > [2] W. Gilpin, “Generative learning for nonlinear dynamics,” Nat Rev Phys, vol. 6, no. 3, pp. 194–206, Mar. 2024, doi: 10.1038/s42254-024-00688-2.
> > >
> > > [3] A. Vaswani et al., “Attention is All you Need,” in Advances in Neural Information Processing Systems, Curran Associates, Inc., 2017
> > >
> > > [4] A. Paszke et al., “Automatic differentiation in PyTorch,” Oct. 2017
> > >
> > > [5] Eisenmann et al. “Bifurcations and loss jumps in RNN training,” Advances in Neural Information Processing Systems, vol. 36, 2024
> > >
> > > [6] A. R. Ives and V. Dakos, “Detecting dynamical changes in nonlinear time series using locally linear state-space models,” Ecosphere, vol. 3, no. 6, p. art58, 2012, doi: 10.1890/ES11-00347.1.
> > >
> > > [7] A. C. Costa et al., “Adaptive, locally linear models of complex dynamics,” Proceedings of the National Academy of Sciences, vol. 116, no. 5, pp. 1501–1510, Jan. 2019, doi: 10.1073/pnas.1813476116.
> > >
> > > [8] O. De Feo and M. Storace, “Piecewise-Linear Identification of Nonlinear Dynamical Systems in View of Their Circuit Implementations,” IEEE Transactions on Circuits and Systems I: Regular Papers, vol. 54, no. 7, pp. 1542–1554, Jul. 2007
> > >
> > > [9] M. Pals et al., “Inferring stochastic low-rank recurrent neural networks from neural data,” Nov. 08, 2024, arXiv: arXiv:2406.16749. doi: 10.48550/arXiv.2406.16749.
> > >
> > > [10] S. Linderman et al., “Bayesian Learning and Inference in Recurrent Switching Linear Dynamical Systems,” in Proceedings of the 20th International Conference on Artificial Intelligence and Statistics, PMLR, Apr. 2017, pp. 914–922
> > >
> > > [11] M. Brenner et al., “Tractable Dendritic RNNs for Reconstructing Nonlinear Dynamical Systems,” in Proceedings of the 39th International Conference on Machine Learning, PMLR, Jun. 2022, pp. 2292–2320
> > >
> > > [12] L. Perko, Differential Equations and Dynamical Systems, vol. 7. in Texts in Applied Mathematics, vol. 7. New York, NY: Springer, 2001. doi: 10.1007/978-1-4613-0003-8.
> > >
> > > [13] J. Guckenheimer and P. Holmes, Nonlinear Oscillations, Dynamical Systems, and Bifurcations of Vector Fields, vol. 42. in Applied Mathematical Sciences, vol. 42. New York, NY: Springer, 1983. doi: 10.1007/978-1-4612-1140-2.
> > >
> > > [14] S. H. Strogatz, Nonlinear Dynamics and Chaos, 0 ed. CRC Press, 2018. doi: 10.1201/9780429492563.
> > >
> > > [15] J. A. Platt et al., “Constraining chaos: Enforcing dynamical invariants in the training of reservoir computers,” Chaos: An Interdisciplinary Journal of Nonlinear Science, vol. 33, no. 10, Art. no. PNNL-SA-191042, Oct. 2023, doi: 10.1063/5.0156999.
> > >
> > > [16] R. Jiang et al., “Training neural operators to preserve invariant measures of chaotic attractors,” Advances in Neural Information Processing Systems, vol. 36, pp. 27645–27669, Dec. 2023.

---

> > > > ### Author Response · Authors · 2024-11-21
> > > > **References (continued)**
> > > >
> > > > [17] Y. Schiff et al., “DySLIM: Dynamics Stable Learning by Invariant Measure for Chaotic Systems,” in Proceedings of the 41st International Conference on Machine Learning, PMLR, Jul. 2024, pp. 43649–43684
> > > >
> > > > [18] Z. Li et al., "Learning chaotic dynamics in dissipative systems," Advances in Neural Information Processing Systems, vol. 35, pp. 16768–16781, 2022.
> > > >
> > > > [19] J. Mikhaeil et al., “On the difficulty of learning chaotic dynamics with RNNs,” Advances in Neural Information Processing Systems, vol. 35, pp. 11297–11312, Dec. 2022.
> > > >
> > > > [20] C. Challu et al., “N-HiTS: Neural Hierarchical Interpolation for Time Series Forecasting,” AAAI 2023.
> > > >
> > > > [21] S. Bai et al., “An Empirical Evaluation of Generic Convolutional and Recurrent Networks for Sequence Modeling,” Apr. 19, 2018, arXiv: arXiv:1803.01271. doi: 10.48550/arXiv.1803.01271.
> > > >
> > > > [22] Vinograd, Amit, et al. "Causal evidence of a line attractor encoding an affective state." Nature 634.8035 (2024): 910-918.
> > > >
> > > > [23] Mante, Valerio, et al. "Context-dependent computation by recurrent dynamics in prefrontal cortex." nature 503.7474 (2013): 78-84
> > > >
> > > > [24] Maheswaranathan, Niru, et al. "Line attractor dynamics in recurrent networks for sentiment classiﬁcation." ICML 2019 Workshop on Identifying and Understanding Deep Learning Phenomena. 2019

---

> > ### Comment · Reviewer_D8qe · 2024-11-22
> >
> > Thank you for the responses, revision, and additional results.
> >
> > Based on your response, my understanding is that DSR aims to learn a generative model capable of producing new data that aligns with the desired summary statistics.
> >
> > Is my understanding correct?
> >
> > If so, I struggle to see how this differs from standard time series (TS) modeling problems. For instance, geometric deep learning models have been applied in scientific domains such as molecular dynamics (MD) simulations or state-space geometry, incorporating similar attractor states as those mentioned in your response.
> >
> > Moreover, modeling the underlying dynamical mechanisms of time series is not a novel topic. Could you clarify what makes this approach or framework stand out? So far, the topic seems well-studied, and the method does not yet appear convincingly novel.
> >
> > I haven’t reviewed the revised manuscript in detail yet, but I wanted to share my preliminary thoughts based on the authors’ response to foster open discussions.
> >
> > Thanks!

---

> > > ### Author Response · Authors · 2024-11-22
> > > **Thank you for your comment**
> > >
> > > Thank you for your reply.
> > >
> > > DSR methods aim to learn a generative model of the time series data, yes, but *long-term statistics* (sometimes called ‘climate statistics’ or invariant measures, i.e. when taking $t \rightarrow \infty$) is not the same as summary statistics. They describe the convergent behavior of the system in the *temporal limit* when unperturbed (see e.g. textbooks by Perko [1] or Guckenheimer & Holmes [2]). Likewise, DSR methods aim to capture the underlying attractor geometry, the closed sets in state space to which *trajectories ultimately converge* (a math. construction), which in our understanding is quite different from geometric deep learning and MD where the question is more about molecular configurations and graph structure. This long-term behavior and state space geometry are key in DSR models, because only those provide some guarantee that the underlying dynamical system was captured. To efficiently achieve this, usually special training techniques (like GTF [3,6]) or loss criteria (based e.g. on Lyapunov spectra [4,5]) are required, which have only been published in the last 2-3 years. We tried to illustrate this in our new Fig. 28, which shows that popular TS models usually do *not* converge to the correct attractor in the time limit, even when equipped with many more parameters.
> > >
> > > But this in itself is indeed *not* the novelty of our paper (while efficient DSR models by themselves and the properties they require are fairly recent, they are indeed established in the field and generally recognized to be different in their aims from TS models [3-7])!
> > >
> > > The novelty is
> > >
> > > 1) that we developed this further into a *hierarchical DSR* approach, which can *simultaneously* capture the dynamics of many different systems living in different dynamical regimes (e.g. new Fig. 27), reconstructing *all of them at the same time* and exploiting the information sampled across these multiple systems for few-shot and transfer learning. It does so highly efficiently, as also witnessed by the surprisingly low computational requirements and scaling, see new Fig. 14; this has not been shown before, at the very least not with this efficiency and strong performance (see Tab. 1),
> > >
> > > 2) we, for the first time to our knowledge, demonstrate that the features extracted from all the DS the hier-shPLRNN sees simultaneously, represent almost 1:1 *control parameters* of the underlying DS. This in our minds is a very surprising and very important observation, because it has fundamental implications for scientific interpretation (like in the cardiovascular example where features are related to biophysical parameters of the underlying DS), and apparently also for downstream time series classification (Fig. 7). This has not been recognized before, and we confirmed this on a large variety of simulated and real-world systems. We view this as the even more important contribution, beyond the architecture or algorithm, because it has so profound implications for scientific ML (like many studies on LLMs, which do not invent a new LLM, but carefully study their properties and capabilities empirically).
> > >
> > > We hope this provides further clarification.
> > >
> > >
> > > [1] L. Perko, Differential Equations and Dynamical Systems, vol. 7. in Texts in Applied Mathematics, vol. 7. New York, NY: Springer, 2001. doi: 10.1007/978-1-4613-0003-8.
> > >
> > > [2] J. Guckenheimer and P. Holmes, Nonlinear Oscillations, Dynamical Systems, and Bifurcations of Vector Fields, vol. 42. in Applied Mathematical Sciences, vol. 42. New York, NY: Springer, 1983. doi: 10.1007/978-1-4612-1140-2.
> > >
> > > [3] F. Hess et al. “Generalized Teacher Forcing for Learning Chaotic Dynamics,” in Proceedings of the 40th International Conference on Machine Learning
> > >
> > > [4] J. A. Platt et al., “Constraining chaos: Enforcing dynamical invariants in the training of reservoir computers,” Chaos: An Interdisciplinary Journal of Nonlinear Science, vol. 33, no. 10, Art. no. PNNL-SA-191042, Oct. 2023, doi: 10.1063/5.0156999.
> > >
> > > [5] R. Jiang et al., “Training neural operators to preserve invariant measures of chaotic attractors,” Advances in Neural Information Processing Systems, vol. 36, pp. 27645–27669, Dec. 2023.
> > >
> > > [6] J. Mikhaeil et al., “On the difficulty of learning chaotic dynamics with RNNs,” Advances in Neural Information Processing Systems, vol. 35, pp. 11297–11312, Dec. 2022.
> > >
> > > [7] N. A. Goering et al.  “Out-of-Domain Generalization in Dynamical Systems Reconstruction,” in Proceedings of the 41st International Conference on Machine Learning

---

> > > > ### Comment · Reviewer_D8qe · 2024-11-22
> > > >
> > > > Thank you for the response, but I remain unconvinced.
> > > >
> > > > Below, I provide two **non-exhaustive examples** to illustrate my concerns:
> > > >
> > > > 1. In geometric deep learning, many invariant measures are time-invariant, such as the symmetry of a system's state or configuration.
> > > > 2. In molecular dynamics (MD) simulations, numerous scenarios focus on energy minima, which serve as attractors for molecular systems.
> > > >
> > > > I appreciate the effort, but I believe the authors need to refine their story further to make it clearer and more compelling.

---

> > > > > ### Author Response · Authors · 2024-11-22
> > > > >
> > > > > We are not contesting that molecular systems may have similar properties (like symmetries or attractors), but this is still very different from systematically assessing agreement in long-term statistics (general invariants in the *time domain*) and attractor geometry (please see Fig. 28 and [1-7]). This in our minds is a much broader and more ambitious goal. For example, systems with an energy functional or potential defined on the whole state space cannot exhibit many dynamical phenomena like chaos (see [1,2] for formal proofs) relevant in all the systems we consider. Molecular configurations fall into some energy minimum (if the referee is referring to protein folding), but there is no long-term dynamics to it (such as in the brain).
> > > > >
> > > > > But this we believe is actually not really the point (we do not want to defend the whole field of DSR here) -- our innovations are specifically those mentioned in points 1-2 above.
> > > > >
> > > > > Is there anything we could do more specifically to convince the referee, or could the referee hint us to other hierarchical DSR models that tackle the same problem or report similar findings, which we may have overlooked?
> > > > >
> > > > > We are happy to dig more into the MD literature and include it in our discussion, but if the referee could more specifically hint us to papers s/he feels are closely related, that would be very helpful. Thanks!

---

> > > > > > ### Author Response · Authors · 2024-11-25
> > > > > >
> > > > > > We wondered whether the referee meanwhile had a chance to look at our actual revisions?
> > > > > >
> > > > > > Our current understanding is that the referee rejects our paper solely on the grounds of a perceived lack in novelty, but so far without hinting us to the specific papers and literature that would support this claim. Further input on this would be much appreciated.
> > > > > >
> > > > > > Please note that we had taken up essentially all your suggestions and addressed them in our revision (like clarifying the differences between TS models and DSR in Fig. 28, or examining other types of TS data in Fig. 27, or the rewrite of sect. 3 etc.).

---

> > > > > > > ### Comment · Reviewer_D8qe · 2024-11-25
> > > > > > >
> > > > > > > I have read your response and revision and decided to maintain my position.
> > > > > > >
> > > > > > > My opinion remains the same as in my previous comment: the authors need to further refine their narrative to make it clearer and more compelling

---

> > > > > > > > ### Author Response · Authors · 2024-11-25
> > > > > > > >
> > > > > > > > We are committed to improving our work and would greatly appreciate the referees' specific suggestions on  how to "refine the narrative". Would it be helpful if we further revised the manuscript and included an additional illustration in the main text to emphasize the crucial aspects of DSR and how it differs from conventional TSA?

---

> > > > > > > > > ### Comment · Reviewer_D8qe · 2024-11-29
> > > > > > > > >
> > > > > > > > > Dear Authors, Chairs and Future Readers,
> > > > > > > > >
> > > > > > > > > In my opinion, this paper is not ready for publication for at least three reasons (there are more). I don't think minor revisions will address these drawbacks in a satisfactory manner.
> > > > > > > > >
> > > > > > > > > 1. The lack of coherence.
> > > > > > > > >
> > > > > > > > >     For example, please see above responses.
> > > > > > > > >
> > > > > > > > >
> > > > > > > > > 2. The draft and responses are not precise and concise.
> > > > > > > > >
> > > > > > > > >     For example, please see above responses.
> > > > > > > > >
> > > > > > > > > 3. The method is not sound.
> > > > > > > > >
> > > > > > > > >     For example, the authors did not adequately address my questions regarding how tractability benefits real science. I don't think the authors' "fundamental implications for scientific interpretation" are truly new or novel. Many feature/factor extraction and explainable AI/ML works have observed and achieved similar results/observations/findings. There are other examples, but I don’t see the point in listing them here.
> > > > > > > > >
> > > > > > > > > In summary, as I pointed out previously, the authors need to present a clear and compelling story to make this work impactful. The current draft is not. There is significant redundancy, inconsistency, contradiction, and inaccuracy in the current draft. Hence, I advocate for rejection, firmly.
> > > > > > > > >
> > > > > > > > > I understand the authors might feel it is urgent to get these results published, but an underdeveloped publication will do more harm than good. I hope you find my suggestions helpful, regardless of the final decision.

---

> > > > > > > > > > ### Author Response · Authors · 2024-11-29
> > > > > > > > > > **Dear Referees, dear Area Chairs, dear Future Readers:**
> > > > > > > > > >
> > > > > > > > > > Dear Referee D8qe, first, please understand we also would like to get the view of the other reviewers, we already got yours and understood you have a strong opinion and rejection bias.
> > > > > > > > > >
> > > > > > > > > > *But please give us a chance to obtain feedback also from the others and do not drown the discussion.*
> > > > > > > > > >
> > > > > > > > > > We are always very open to constructive feedback and specific suggestions on how to improve our paper. However, your feedback is not constructive, not specific enough to act upon, gives no hints to the relevant literature, no evidence to support your points, even after explicitly asking for it multiple times.
> > > > > > > > > >
> > > > > > > > > > Your latest reply is a good example:
> > > > > > > > > >
> > > > > > > > > > 1) Please specify what you mean, see our replies above.
> > > > > > > > > >
> > > > > > > > > > 2) The points you raised we specifically replied to, but many of your points were just too unspecific or general, without a single literature reference. In these cases we asked for further specification, but you gave no answer.
> > > > > > > > > >
> > > > > > > > > > 3) Tractability is obviously important for science, obtaining understanding and insight into mechanisms is at the core of the scientific process. With regards to DSR it has been discussed at length in the literature (please have a look at the many papers we had cited in our previous responses, if you are not so familiar with the lit.). Many approaches in dynamical systems reconstruction (see for instance all the work from the Brunton group, or papers by Ostijic or Sussillo in neuroscience) is about nothing else than understanding the dynamical mechanisms, attractor objects, convergence properties etc. of the underlying system.
> > > > > > > > > > You claim there is much work which has provided similar observations – then please provide specific citations.
> > > > > > > > > >
> > > > > > > > > > If you honestly want to be helpful, then please let us specifically know where you see the redundancies, the contradictions, the inaccuracies and the inconsistencies in our draft, because none were mentioned in your initial response (or were easy to refute). Please try to be specific and give us *specific* examples (because in your comments above there are none), and please give us *specific* pointers to the literature you have in mind.
> > > > > > > > > >
> > > > > > > > > > Dear other referees, we are still interested in your take on this, and your views on our revision. Thank you.

---

> ### Comment · Reviewer_D8qe · 2024-11-30
>
> I am sorry that the authors find my comments not "honestly helpful." However, I am not sure I appreciate the the authors' tone.
>
> My critique is precise. **The authors had the opportunity to provide a rebuttal to improve their score (which I was open to considering), but their response failed due to its excessive length, inconsistency, and logical gaps.** Unless these issues are resolved, it is not productive for me to provide specific examples or citations to guide the authors. Repeated inquiries from the authors did not and can not change this fact. It is the authors' responsibility to address these problems and clarify their work. It is not the reviewer's duty to guide the authors' research.
>
> My only reason for posting my previous response was that the authors asked for clarification again. It is weird that the authors are now accusing me of "drowning the discussion." I find this behavior highly unprofessional. For this, I am lowering my score to a strong reject.

---

> > ### Comment · Reviewer_Qfhy · 2024-11-30
> > **I disagree, and I do not think this is fair**
> >
> > Hi Reviewer D8qe, I think that this is a bit disproportionate. I've read through the thread, and I understand your dissenting opinion regarding novelty and clarity, though I personally disagree on these points. However, I think the authors have appropriately aimed to use the rebuttal period to rebut your conceptual concerns, as they should be expected to do. Your concerns primarily address the fundamental topic of the paper, and so the authors have few options to respond to you other than to disagree, and so it's appropriate for them to push back.
> >
> > Moreover, I think that your recent comments have been needlessly negative towards the authors.
> >
> > Regarding the scientific content, in my opinion dynamical systems reconstruction is a well-established field, and it is not the authors' responsibility to justify this entire field, which is a mature and active area of research. I don't feel that the authors should have to compare their work on physiology datasets to unrelated works on molecular dynamics or other subfields irrelevant to their paper's topic.

---

### Official Review · Reviewer_vh2v · 2024-11-04

**Soundness:** 3
**Presentation:** 4
**Contribution:** 3
**Rating:** 8
**Confidence:** 4

**Summary:**

The paper presents a method for learning dynamical systems by leveraging information from multiple observations (referred to as ``subjects'') for scientific analysis. The authors achieve this by learning shared parameters across observations while preserving individual variability through a hierarchical model for time-series and dynamical systems reconstruction. They validate their approach using both synthetic data (e.g., Lorenz attractor) and real-world data (e.g., EEG), demonstrating that their method captures unique dynamical characteristics that other methods overlook.

**Strengths:**

I really enjoyed reading the paper and believe it addresses an important issue in computational neuroscience, where the data often includes multiple non-simultaneous observations across subjects. Moreover, the paper is well written, and the results clearly underscore the model's capability compared to other methods. They demonstrate their approach on multiple examples, including both synthetic and real-world datasets, under diverse settings.

**Weaknesses:**

# Critical:
1) While the paper was overall very well written, I believe that the Related Work (DSR section) missed discussion about DSR methods based on time-changing combinations of linear dynamical operators, including both  switching, e.g., [1], and decomposition [2] models.
2) I also missed from the Related Work more methods that are designed or capable of leveraging multi-session information in neuroscience specifically, including both [3,4]. I believe that these should appear in the introduction or towards the end of the related work, or even be benchmarked against if possible (although an explanation of why these methods are insufficient may suffice).
3) I think an important question that needs to be discussed is how many subjects you need for the method to succeed. For instance, if you have a total of $J$ subjects, what will happen if you leverage information from $J-1$ or$ J-2$? I would assume that performance will improve rapidly with low $J$ values and then level off as $J$ increases to higher numbers, potentially resembling a logarithmic function.
4) A main concern of mine is the need for a discussion on the minimum overlap required between subjects in terms of dynamical modalities, channels, and time scales for the model to work effectively. While the authors show that different dynamical regimes can be modeled in synthetic data (e.g., through varying aspects of the Lorenz attractor under changing parameters), I assume that data from completely distinct dynamical processes—such as rapid interactions in neuropixels data combined with annual climate changes—or recordings from unrelated units even within the same field (e.g., neural activity from very different brain regions with zero overlap across subjects) might challenge the model’s effectiveness. I suspect there is a critical level of cross-subject similarity needed in both scales and channel identity for reliable performance. A complete benchmark may be impractical due to the sheer range of possibilities, but providing a list of mathematical assumptions or briefly addressing this in the discussion/limitations section would clarify these requirements.
5) The model requires a lot of hyperparameters.

# Minor but recommended:
1) Line 67: I believe an em dash (--- in LaTeX) should be used here instead of an en dash (-).
2) I would recommend that the authors more clearly state that the term `group` refers  to the observations from all subjects. Initially, I was unsure if `group` referred to some internal grouping within the data, so a brief clarification the first time you use it would help.
3) While I understand the page limit, it does not seem sensible to place the illustration figure (Fig. 7) in the supplementary materials. This figure is important for understanding the model, and the authors would want readers to see it. I suggest making it shorter and moving it to the main text.
4) Line 171-172: I suggest $P_{w_1} \in \ldots $ should be kept on a single line (i.e., by wrapping that formula in curly braces).
5) It is unclear how robust the method is to different model initializations.
6) The colormap in Fig. 4a seems off: the ticklabels do not align with the color change. I suggest using a legend with `Subject 1`, `Subject 2`, etc., instead of a colorbar, as that should be categorical, not sequential (unless Subject 1 is indeed more similar to Subject 2).


[1] Linderman, S., Johnson, M., Miller, A., Adams, R., Blei, D., & Paninski, L. (2017, April). Bayesian learning and inference in recurrent switching linear dynamical systems. In Artificial intelligence and statistics (pp. 914-922). PMLR.

[2] Mudrik, N., Chen, Y., Yezerets, E., Rozell, C. J., & Charles, A. S. (2024). Decomposed linear dynamical systems (dlds) for learning the latent components of neural dynamics. Journal of Machine Learning Research, 25(59), 1-44.

[3] Pandarinath, C., O’Shea, D.J., Collins, J. et al. Inferring single-trial neural population dynamics using sequential auto-encoders. Nat Methods 15, 805–815 (2018). https://doi.org/10.1038/s41592-018-0109-9

[4]  Mudrik, N., Ly, R., Ruebel, O., & Charles, A. S. (2024). Creimbo: Cross ensemble interactions in multi-view brain observations. arXiv preprint arXiv:2405.17395.‏

**Questions:**

1) Regarding Eq. 4, in line 296 you mentioned for the first time that it is piecewise-linear. Did you mean it is locally linear? Or are there full periods of linearity that switch between them (e.g., as in [1] from my Weaknesses text cell)? From equation 4 itself it just seems linear so  I would clarify it near the equation itself if that is the case.
2) Can you give some intuition for the meaning of the different $ P $s (e.g., $P_A$, etc.)? I would interpret them as capturing groups of features or parameters that operate together to link the latent and observed space.
3) In line 185, why did you choose to train the model directly on the observations  (i.e., the $x_t = z_t$ assumption)? What do you think will be the effect if a different approach is chosen?
4) When describing the batching, based on the notations, it seems that all samples start with the first time point (due to writing $x_1 \ldots $ in line 191). Is this correct, or is it a mistake and batches taken from some random $\tau$ to $\tau + {T_{\text{seq}}}$?
5)  What does `2X10` refer to in row 495? Do you mean 20 regions? How much overlap is there between them?
6)  How would you address temporal nonstationarities within your model?
7) What is the model's computational complexity? How does it increase with the number of subjects?

---

> ### Author Response · Authors · 2024-11-21
> **point-by-point reply Weaknesses**
>
> We thank the referee for the very supportive and encouraging feedback!
>
> **Critical Weaknesses**
>
> *W1 (Related Work on DSR)*: Thank you for pointing out these references, which we now have integrated into our discussion of related work.
>
> *W2 (Related Work on Hierarchical Modeling)*: Thank you, we now mention these in the 2nd pg. of our related work sect.. LFADS, however, in our perception is not strictly designed for DSR but more for posterior inference. Volkmann et al. [5], for instance, show LFADS is not capable of reconstructing dynamical invariants, like attractor geometries, from fMRI data. Combining data from multiple sessions is also not quite the same as reconstructing dynamics from many different systems and dynamical regimes: While LFADS allows for session-specific en-/decoders to accommodate, e.g., different neuron numbers, as far as we can see it does not have subject-specific parameters for the recursion model that would allow for reproducing subject-specific regimes, but the generative model and dynamics are assumed to be the same across sessions. The other paper [4] seems very recent and currently does not provide any code which would allow for a direct comparison (to us it appears it is under review for this same conference). Hence, both methods are difficult to directly compare with (we did, however, include a number of other new comparisons in the revision, see list of additional experiments in general response).
>
> *W3 (dependence on number of subjects)*: Yes, fair point! We now systematically tested this, both as a function of the number of subjects and the length of the time series, see new pg. ‘Scaling and robustness’ at end of sect. 4.1 and new Fig. 13. The referee’s intuition is right, as the number of subjects $S$ increases, performance gains are initially strong and then start to level off logarithmically. Importantly, however, multi-subject information *always* helps to improve performance, and very profoundly so when the time series are rather short (see also Fig. 12).
>
> *W4 (overlap between subjects required)*: Thank you for bringing this up. First, please note that in the experiments on the cardiovascular, the EEG or the fMRI data, indeed many different subjects, conditions, or brain regions were combined, with, in part, largely different dynamics (e.g. epileptic activity vs. normal EEG), showing that this is not only possible, but rather a strength of our approach. We would argue that these differences in dynamical regime (e.g. chaotic vs. periodic) that our model is capable of integrating (as also systematically explored in the simulated benchmarks), is the more crucial aspect, rather than the different systems (subjects) the time series come from.
>
> To further address the referee’s point, we have now added to our revision new experiments where the hierarchical model simultaneously reconstructed dynamics from *three completely different systems* (Lorenz, Rössler and Chua system). We found that still only low-dimensional ($\leq 4$) feature vectors were required to faithfully reproduce dynamics from all three systems and the different dynamical regimes, with a semantically clearly interpretable feature space, see new Fig. 27 and end of revised pg. on ‘Multidimensional parameter spaces’ (sect. 4.2).
>
> Of course, at some point, if dynamics from vastly different systems evolving on completely different time scales are to be combined (like the referee’s example of neuropixels vs. annual climate data), our approach would likely need to assign rather high-dimensional feature vectors to still accommodate these differences. On the other hand, from a practical perspective, this may also yield a recipe for how to use our framework to decide how dynamically related different systems are: If they occupy largely orthogonal regions in the model’s state and feature spaces, this would indicate they are dynamically rather unrelated. We have included these thoughts in our revised Discussion.
>
> *W5 (number of hyper-parameters)*: Not really, probably our presentation has evoked a bit of a wrong perception here: On top of the usual parameters one would need to specify in any deep learning framework (like model size L), only the size of the subject-specific feature vectors needs to be provided, as now clarified in the new pg. at the end of sect. 3.2 (see also Appx. A.1.11 and Table 5 for more details). The GTF rate $\alpha$ can in fact also be determined automatically (called ‘adaptive GTF’, see Hess et al. [6]), but for the range of datasets we explored, simply adopting default settings from [6] worked (i.e. w/o further grid search). In fact, all-in-all, there were only two hyper-parameters which needed tuning.

---

> > ### Author Response · Authors · 2024-11-21
> > **Minor Weaknesses \& Questions**
> >
> > **Minor points**
> >
> > *M1,2,4,6*: Thank you for pointing out, all of these were now corrected/ amended as suggested.
> >
> > *M3*: We understand the referee’s point, and admit that moving Fig. 7 to the Appx. in favor of more data results was not a ‘light-hearted’ decision (even more pressing now with the many new results produced for the rebuttal). But we reduced it in size and moved it back to the main text now!
> >
> > *M5*: Good point. Since all results reported were always across 10 training runs, we have this information and in the revision now report several statistics quantifying the between-run agreement (correlations between feature vectors and agreement in attractor geometry and power spectra across runs), see new pg. ‘Scaling and robustness’ at end of sect. 4.1, Appx. A.1.5 and new Fig. 11.
> >
> > **Questions**
> >
> > *Q1 (piecewise-linear)*: Yes this could have been clearer. Note that in the previous version of the ms. eq. 4 is the same as eq. 3, only with a superscript j to highlight the subject-specificity. Both eqs. contain the piecewise-linear (thus nonlinear) term $\phi(z)=max(z,0)$. One could see this as a form of ‘locally linear’, but in the math. and dynamical systems lit. piecewise-linear (PWL) is the more common term for functions that are defined on a domain segmented into pieces (regions) with different linear behavior (see e.g. [7]). This is indeed similar to [1], only that in our case the PWL functions are continuous across the boundaries. In our revision, we now combined eq. 3 \& 4 and clarified this point directly below eq. 4.
> >
> > *Q2 ($\mathbf{P}$ matrices)*: The $\mathbf{P}$’s are group-level matrices (shared across all subjects) that, when combined with the subject-specific features, give rise to the model parameters indicated in the respective subscripts. Thus, $\mathbf{P}\_{\mathbf{A}}$ multiplied by the feature vector for subject $j$ gives rise to the subject-specific model instantiation of matrix $\mathbf{A}$ in eq. 4,  $\mathbf{P}_{\mathbf{W}_1}$ multiplied by the subject-specific feature vector to $\mathbf{W}_1$ in eq. 4, and so forth. So these matrices capture the common group-level properties of the dynamics, which yield a subject-specific model instantiation when combined with the subject-specific feature vectors. We have thoroughly rewritten the whole sect. 3.1 (and also added a new illustration, Fig. 8) and hope all this is much clearer now.
> >
> > *Q3 (observation model)*: This is usually done, following recommendations elsewhere in the DSR literature (Hess et al. [6]), to put the burden on capturing the actual dynamics onto the latent model. A more expressive observation (decoder) model incurs the risk that too much of the job of accounting for the actual observations is taken over by the (static) decoder (this may be seen as a type of over-fitting), instead of by the latent dynamics as desired. The identity mapping between the latent space and the observations was chosen to minimize this risk. We now mention this in the third pg. of sect. 3.1.
> >
> > However, we now also repeated part of the experiments with more expressive decoder models, with a linear operator $\mathbf{B}$ mapping from the latents to the observations in eq. 2, where $\mathbf{B}$ was either taken to be subject-specific, $\mathbf{B}^{(j)}$, or common across subjects (see Appx. A.1.6). While results remained fundamentally the same (see Table 3), performance decreases slightly.
> >
> > *Q4 (batching notation)*: $T_{max}$ denotes the total length of the time series for each subject. During training, we do not use the entire sequence at once. Instead, we randomly sample subsequences of length  $T_{seq}$. These subsequences are drawn from the full sequence starting, indeed, at a random index $T_{start} \in [1,T_{max}−T_{seq}+1]$. We apologize for the confusion, and hope this is now described more clearly in Appx. A.1.4 (we moved the details of the batching to the Appx., first, to make this clear and, second, because we needed additional space in the revised main text for all the new material).
> >
> > *Q5 (brain regions)*: Yes, apologies for not making clear enough, 20 non-overlapping regions in total, with 10 from each lateral hemisphere (now clarified). In Appx. A.3 we furthermore added information about the specific brain regions, which span cortical areas related to working memory processing, mainly prefrontal and parietal cortical areas.

---

> > > ### Author Response · Authors · 2024-11-21
> > > **Questions (continued) & References**
> > >
> > > *Q6 (nonstationarity)*: Good point. Just for clarity: Under non-stationarity we understand a systematic shift in the statistical moments of the distribution over time series observations across the whole observed period (like a trend in mean or variance).
> > > The model itself is completely agnostic about whether the time series are stationary or not in this sense, i.e. can accommodate non-stationarity through dimensions in its latent space (similar to the way each non-autonomous dynamical system can math. always be rewritten as an autonomous one, e.g. Strogatz [8]; see also new Fig. 26 where the data are non-stationary). It is more a matter of model interpretation and data preprocessing, i.e. whether the non-stationarity is of interest itself or should better be regressed out beforehand (like a trend due to electrode drift).
> > >
> > > One interesting strategy and use case for our model is actually splitting non-stationary time series into piecewise stationary segments across ‘subjects’ (systems), which in essence reflects the situation we tested with our benchmarks (with different time series created by ground truth models with different parameters, thus differing in their statistics): In that case, the subject features would acquire the meaning of control parameters that underlie the system’s non-stationary evolution, thus potentially enable to identify drivers of the non-autonomous behavior. We added a remark on this potential application at the end of the revised Conclusions sect. (before ‘Limitations’).
> > >
> > > *Q7 (comp. complexity)*: To answer this question, in new Fig. 14a we evaluated the total training time until a specific pre-set performance level was reached ($D_{stsp} \leq 1.0$) on a single CPU core as a function of the number of subjects $S$: To our own surprise, counterintuitively, median computation time in fact *decreases* with $S$. We believe the reason for this is that with more subjects, the hier-shPLRNN has access to a larger portion of the underlying system’s parameter space, and can efficiently leverage this information to speed up training and reconstruction on each single subject. Hence, the pre-set performance criterion could be reached much earlier, offsetting the increase in $S$ and testifying to the efficiency of our approach in integrating and transferring information across multiple subjects.  The pure *inference time* for a time series of fixed length $T_{max}$ scales indeed linearly with $S$ and the number of features $N_{feat}$ (formally, and as confirmed in new Fig. 14b,c).
> > >
> > > **References**
> > >
> > > [1] Linderman, S., Johnson, M., Miller, A., Adams, R., Blei, D., & Paninski, L. (2017, April). Bayesian learning and inference in recurrent switching linear dynamical systems. In Artificial intelligence and statistics (pp. 914-922). PMLR.
> > >
> > > [2] Mudrik, N., Chen, Y., Yezerets, E., Rozell, C. J., & Charles, A. S. (2024). Decomposed linear dynamical systems (dlds) for learning the latent components of neural dynamics. Journal of Machine Learning Research, 25(59), 1-44.
> > >
> > > [3] Pandarinath, C., O’Shea, D.J., Collins, J. et al. Inferring single-trial neural population dynamics using sequential auto-encoders. Nat Methods 15, 805–815 (2018). https://doi.org/10.1038/s41592-018-0109-9
> > >
> > > [4] Mudrik, N., Ly, R., Ruebel, O., & Charles, A. S. (2024). Creimbo: Cross ensemble interactions in multi-view brain observations. arXiv preprint arXiv:2405.17395.‏
> > >
> > > [5] E. Volkmann et al.  “A scalable generative model for dynamical system reconstruction from neuroimaging data,” Nov. 05, 2024, arXiv: arXiv:2411.02949. doi: 10.48550/arXiv.2411.02949.
> > >
> > > [6] F. Hess et al. “Generalized Teacher Forcing for Learning Chaotic Dynamics,” in Proceedings of the 40th International Conference on Machine Learning, PMLR, Jul. 2023, pp. 13017–13049.
> > >
> > > [7] V. Avrutin et al., Continuous And Discontinuous Piecewise-smooth One-dimensional Maps: Invariant Sets And Bifurcation Structures. World Scientific, 2019.
> > >
> > > [8] S. H. Strogatz, Nonlinear Dynamics and Chaos, 0 ed. CRC Press, 2018. doi: 10.1201/9780429492563.

---

> > > > ### Author Response · Authors · 2024-11-29
> > > > **any comments?**
> > > >
> > > > Dear Referee vh2v,
> > > >
> > > > We wondered whether you had a chance to look at our extensive revisions, and whether you would have any further feedback or remaining questions?
> > > >
> > > > Thank you!

---

> ### Comment · Reviewer_vh2v · 2024-11-30
> **response to authors**
>
> First, I sincerely apologize for my late reply. I read the authors' comments early on and wanted to ensure I had the proper time to thoroughly review their additions and explanations. Please know that the delay does not reflect a lack of interest nor a lack of appreciation for your work and rebuttal efforts.
>
> ### Regarding the authors' responses:
>
> **W2:** I understand and agree with the comments about LFADS and [4]. While I still think LFADS is relevant to mention, I recognize the differences. Also, the [4] I referenced seems to be from May.
>
> **W3:** I appreciate the additional experiments, as they clarify many aspects!
>
> **W4:** I think the exact clarification you provided in your response to W4 would be a valuable addition to the discussion in your paper.
>
> **M3:** While I am sure including the figure in the appendix was not a light decision, I maintain that it is crucial to have it in the main text, as you have now done.
>
> **Q6:**  I understand the authors' point about non-stationarity. However, in real-world applications, particularly with biomedical data, non-stationarity is a critical consideration. Equation 4 is piecewise linear, implying an assumption that the main non-stationarity arises from abrupt system behavior changes. While this is a common assumption in recent works (e.g., [1, 2]), it may not align well with how biological data are typically assumed to evolve. I recognize that this is not the central focus of the paper and want to clarify that I do not expect the authors to address this experimentally or modify their dynamic equation model within the scope of this version of the paper. However, given the importance of non-stationarity, particularly in biomedical contexts, I believe it would be valuable to discuss this in the paper’s discussion section or acknowledge it as a limitation or a direction for future work.
>
>
>
> **I am still reading and reviewing the new experiments and additions and may provide additional comments by the end of the discussion period. Thank you again for the responses.**
>
> [1] Zhang, Y., & Saxena, S. Inference of Neural Dynamics Using Switching Recurrent Neural Networks. In The Thirty-eighth Annual Conference on Neural Information Processing Systems.
>
> [2] Karniol-Tambour, O., Zoltowski, D. M., Diamanti, E. M., Pinto, L., Brody, C. D., Tank, D. W., & Pillow, J. W. Modeling state-dependent communication between brain regions with switching nonlinear dynamical systems. In The Twelfth International Conference on Learning Representations.
>
> [3] Mohammadi, Z., Ashwood, Z. C., International Brain Laboratory, & Pillow, J. W. (2024). Identifying the factors governing internal state switches during nonstationary sensory decision-making. bioRxiv, 2024-02.

---

> > ### Author Response · Authors · 2024-11-30
> >
> > Thank you very much for your kind response!
> >
> > **W2:** Yes, agreed, and note we did indeed include both papers in our Related Work section.
> >
> > **W4:** Thank you, agreed as well; W4 we had addressed partly by incl. new Fig. 27 and the last pg. of the sect. on ‘Multidimensional parameter spaces’, and partly by adding discussion in Limitations. But we are happy to expand further on this and add the clarifications in the first pg. of our W4 response above to the appropriate sub-sect. in 4.2 \& 4.4. We will make room for this through the modifications suggested below at the end of our response to Q6.
> >
> > **Q6:** Yes, absolutely, we completely agree non-stationarity is an important issue in real biomedical, physiological and many other applications (we know first-hand!), apologies if our response gave a wrong impression here. We just wanted to point out that our hierarchical model offers interesting new possibilities here for actually inferring the drivers of non-stationarity (as briefly noted now at the end of Conclusions, before Limitations).
> >
> > As stated in our previous response, this type of model (even without hierarchization) can perfectly capture non-stationarity in its latent space (examples of non-stationarity were in Fig. 26, but also in the EEG data). But if we understood the referee correctly, the concern is more that this non-stationarity is captured the “wrong way” (rather through an abrupt jump), due to the ReLU? Although a reasonable worry, this is in fact usually not the case, the model would still predict a smooth transition, if the underlying transition in the data is smooth (which the model tries to capture). Note, for instance, that all the benchmarks used (like the chaotic systems) are smooth, continuous-time dynamical systems, yet they are captured through combinations of ReLUs/ piecewise-linear functions (note the ReLU itself is also a continuous function, although not continuous in its first derivative at 0 of course). But the more important point to mention here may be that ReLU-based networks can indeed be universal approximators of smooth functions (e.g. [1-3]), despite the discontinuity in the first derivative.
> >
> > We agree with the referee that this topic may in any case deserve more attention in our final section. We will therefore include an explanation along the lines above, and add another illustration in the Appx. showing how the non-stationarity in the real data is represented within the model’s latent space (note we can no longer directly update our manuscript at this point). One possibility to make room for this is to move the pg. on fMRI data entirely to the Appx., another would be to move the second pg. in the EEG sect. to the Appx. (recommendations on this are welcome).
> >
> > We are also happy to include the additional references noted by the Referee in our Related Work sect.
> >
> > [1] Huang, Changcun. (2020). ReLU Networks Are Universal Approximators via Piecewise Linear or Constant Functions. Neural Computation. 1-30. 10.1162/neco_a_01316.
> >
> > [2] Lu, Zhou and Pu, Hongming and Wang, Feicheng and Hu, Zhiqiang and Wang, Liwei (2017). The Expressive Power of Neural Networks: A View from the Width. Advances in Neural Information Processing Systems 30.
> >
> > [3] Linhao Song, Ying Liu, Jun Fan, Ding-Xuan Zhou (2023) Approximation of smooth functionals using deep ReLU networks, Neural Networks, Volume 166, Pages 424-436.

---

> > > ### Comment · Reviewer_vh2v · 2024-11-30
> > > **response to authors**
> > >
> > > **I thank the authors for their quick response.**
> > >
> > > To clarify the last point about non-stationarity, I did understand that the current modeling choice is expressive enough to capture non-stationarities in general. However, the ReLU function imposes constraints on how it models the non-stationarity by limiting it to follow the ReLU (max(0, value)) constraint. These constraints are not specifically tailored to capturing the internal changes in co-occurring non-stationary time-varying biological processes in an interpretable and biologically aligned way.
> > >
> > > For example, consider the case of the fMRI data you used, collected during cognitive tasks. There are likely multiple co-occurring processes, e.g., memory encoding, stimulus-related processing, and visual processing, that evolve together in a non-stationary manner. These processes may drive latent brain interactions that change over time in ways that are not necessarily best described when constrained to ReLU activations. Instead, the internal interactions themselves may present temporal changes relative to each other. This suggests that transition matrices should ideally vary over time (beyond ReLU / piecewise) to extract such hidden dynamics and better link them to how the brain (or other biomedical data) evolves.
> > >
> > > That said, modeling transitions in a way that changes all the time with alignment to biology, while certain works I mentioned in my first response try to address that, is still not trivial and can lead to less robust solutions. This remains a significant challenge in dynamical systems modeling in science, and **I do not expect the authors to solve it in this paper**. Particularly, I acknowledge this is not the primary focus of the paper and do not expect the authors to change their modeling choice for that.
> > >
> > > **However, I do believe it is important to note in the paper that this is a modeling choice, which—naturally—has limitations, particularly in constraining the ability to capture time-varying internal changes in biological processes in an interpretable and biologically aligned way, and to consider this for future work.**
> > >
> > > I raised my score to 7 as I believe that the rebuttal changes and additions improve the paper.

---

> > > > ### Comment · Reviewer_vh2v · 2024-11-30
> > > >
> > > > P.S. I realized that a rating of 7 is not an option in this venue. Please allow me more time to review the additions and decide whether to give an 8 or keep the 6. **For the record and for the AC's attention, I believe it should be a 7.**

---

> > > > ### Author Response · Authors · 2024-12-01
> > > > **Thanks for clarifying**
> > > >
> > > > Thank you for clarifying, we now understood the referee’s point. Yes, in the biological substrate one may assume, for instance, that the underlying connectivity itself adapts to changing contingencies or task requirements. Our system, as formulated in the current version of the paper (but please see below), would likely represent this process through other changes in its latent space, and we completely agree this is a limitation in terms of biological interpretability. How about a paragraph like this in the final draft (under Limitations):
> > > >
> > > > “While we demonstrated that our hierarchical framework offers a level of interpretability by relating subject features to underlying control parameters, of course, as with any ML/AI modeling choice, there are naturally also limitations w.r.t. interpretability. For instance, in neuroscience, one important source of non-stationarity are functional changes in the underlying network connectivity that may occur across the course of an experiment. A generic DSR model like ours would likely capture these through slow changes in its latent space, but is not set up to model the process of weight plasticity itself (as, e.g., in [1-3]), leaving room for future extensions.”
> > > > … where we would cite in this context the references provided by the referee.
> > > >
> > > > On top, however, we now have run new experiments where we concretized the idea we mentioned in our initial response to your comments: We generated a time series with a time-varying control parameter $\rho \in [28,68]$ for the Lorenz-63, simulating a setting where control parameters slowly drift. We then co-trained an additional function, parameterized by an MLP, that maps time and the current subject-specific feature vector onto a time-dependent feature vector. This resulted in an explicitly time dependent DSR model with a time-varying connectivity matrix, as suggested by the referee. In preliminary experiments, this allowed us to almost perfectly capture the temporal drift in $\rho$ in the feature value (linear regression $R^2=0.98$). This provides a proof of principle that a slight amendment of our setting may also be used to indeed represent non-stationarity in a biologically more interpretable way. We are currently running further experiments on this which we will include in our final revision.

---

> > > > > ### Comment · Reviewer_vh2v · 2024-12-03
> > > > >
> > > > > I thank the authors for their reply. As I believe that the rebuttal additions improve the paper and enjoyed our discussion above on non-stationarities, I will raise my score to 8.

---

> > > > > > ### Author Response · Authors · 2024-12-03
> > > > > >
> > > > > > We thank the referee very much for the great feedback, and for engaging in such a fruitful discussion with us.

---

### Official Review · Reviewer_LBGT · 2024-11-04

**Soundness:** 3
**Presentation:** 3
**Contribution:** 3
**Rating:** 6
**Confidence:** 3

**Summary:**

This paper tackles the problem of analyzing multiple time series to extract the underlying dynamical properties that govern the observed data.  The authors propose hierarchical variant of the shallow PLRNN model which introduces group-level parameters in a subject latent space.  They propose a training scheme and implement/test the method on various experiments involving transfer learning and few-shot learning, comparing their method's performance with individual time series models and other models in the literature.  They also perform various analyses to showcase how their method is interpretable and uncovers ground-truth information about the underlying dynamics.

**Strengths:**

- This was a very nice paper to read, well-organized and well-written.  The authors included an extensive related works section to cover many developments related to their work.  In general, the presentation was quite clear, the figures were nicely formatted, and the experimental details were explained well.
- The problem of extracting dynamical insights from many time series is a problem of both established and burgeoning interest across many fields -- particularly in scientific ones in which data is plentiful, yet we believe there are underlying laws that determine how the data is generated.  The ability to incorporate group-level information / multi-task learning into a model arguably has the potential to learn these dynamics better; it is exciting that the authors are able to show notable improvements in performance over individual-level models and some recently proposed other methods.
- I appreciate the authors' analyses to provide further insight into why their method is able to perform well.  It is one thing to obtain better numerical performance on a task; it is another to perform well for the right reasons.  The interpretability experiments are interesting because they convey that the model has been able to capture some knowledge of the underlying physics, which is something that one cares about in scientific applications.
- The authors perform several experiments to showcase their method in different settings.  It appears that the method would have wide applicability to a range of different problems.
- The authors provide a link to their code and the code appears well-organized and a useful resource for the community.

**Weaknesses:**

- The model development in a bit limited.  While the authors mention that their method is technically applicable to any DSR or TS model, they mainly focus on shPLRNN.  Their experiments focus on this one instantiation; it would be interesting to consider how the hierarchization could be applied to other models as well.
- The training method is also an amalgamation of many established techniques in the literature.  Though there is limited development of any one of these individual techniques, it should be noted that combining many ideas and getting things to "work" in practice is also a useful contribution.

**Questions:**

- In Table 1, is there any sense of how your results vary as a function of T_max?  I'm wondering if there's a substantial regime of T_max (i.e. time series length) in which the group model performs much better than single time series models?  It would be interesting to see at what point performance between group/single becomes similar, if at all.  This could give practitioners a sense of what regimes the group model has an advantage over individual level models.
- As the authors mentioned in their literature review, there is a large body of work on sparse identification of nonlinear dynamics (i.e. Sindy) and physics-informed neural networks (PINNs) -- alternative approaches that have tackled some of the same datasets (e.g. Lorenz) as those used in this paper.  Is there a sense of how well your method compares against these other approaches in the literature (e.g. in an experiment such as Section 4.1)?

---

> ### Author Response · Authors · 2024-11-21
> **point-by-point reply**
>
> We thank the referee very much for the valuable comments and appreciation of our work!
>
> **Weaknesses**
>
> *W1 (applic. to other models)*: This is a fair point. We chose the shPLRNN because it’s a SOTA in the field [2], but now have repeated some of the experiments with vanilla RNN (trained by GTF as in [2]), LSTM (as in Vlachas et al. [4]), and Markov Neural Operators (Li et al. [5]), to highlight the generality of our approach; see reworked sect 3.1 (below eq. 4), Appx. A 1.6, Appx. A.5.1, new Table 3, and new Fig. 17.
>
> *W2 (contributions)*: Thank you for acknowledging that while, indeed, our approach built on previous work, it required many components and a specific construction for the group- and subject-level parameters to work as successfully as presented here. We view this and the demonstration of the efficacy of the approach as one major contribution. Another one is the observation that the subject feature spaces had a clear semantic structure with an interpretable relation between features and crucial control parameters of the underlying system that has not been reported before to our knowledge, but is important for scientific or medical applications.
>
> **Questions**
>
> *Q1 (scaling with $T_{max}$)*: This is a good question, which we had addressed in Appx. A.1.8, Fig. 12 (formerly Fig. 10), where we investigated relative performance (hierarchical/baseline) as a function of time series length for the 64-subject Lorenz-63 dataset. As this figure (and new Fig. 13) show, increases in performance are essentially obtained for all time series lengths, but are particularly profound for $T_{max} \leq 500$. Admittedly this point was a bit lost in the main text, and we made it more prominent now in a new pg. on ‘Scaling and robustness’ at the end of sect. 4.1. In addition, we now also studied more systematically how performance depends on both, number of subjects and time series length, see new Fig. 13.
>
> *Q2 (SINDy \& PINNs)*: Both SINDy and PINNs are different from our approach in the sense that they both require specific prior knowledge about the functional form of the presumably underlying ODE system. In several previous publications where SINDy was pitched against other approaches, including the shPLRNN used here as a basic reconstruction model (see [2]), SINDy performed comparably *as long as the pre-specified function library matches the ground truth (GT) system* (i.e., contains the same terms, like multinomials of correct order for the Lorenz-63). But it breaks down as soon as the function terms describing the GT are lacking from its library [1]. In particular, SINDy failed on essentially all real-world datasets checked in [2], as for those sufficient prior knowledge is often not available (and using larger libraries is computationally infeasible, as we now explicitly confirmed for all three real-world applications considered in our paper, where even just numerically it became impossible to train SINDy on the complete set of (high-dimensional) observations and trained models always diverged). This is a crucial point of course, as at the end of the day we are primarily interested in those real-world applications where sufficient priors may not be available.
>
> For PINNs, similarly, prior physical knowledge about the functional form of the presumably underlying ODE system is required (implying somewhat similar issues as with SINDy, e.g. Krishnapriyan et al. [3]). In general, the prior should also be sufficiently broad to accommodate *all* systems trained on, not just one particular system. One such more generic prior could be strong dissipativity in a class of DS. This assumption is used in Markov Neural Operators [4], and so we now tested MNOs as an architecture with a physics-inspired regularization loss within our hierarchisation framework (Appx. A.5.1); new Fig. 17 essentially reproduces the observations from Fig. 3 with the MNO, likewise obtaining an about linear relation between the inferred feature value and the true control parameter.
>
> Thus, incorporation of physical priors as in PINNs is fully compatible with our framework (and our new experiments suggest won’t change the results). But it may require more detailed study in its own right, as we now also commented on at the end of our Conclusions sect. (in Limitations).

---

> > ### Author Response · Authors · 2024-11-21
> > **References**
> >
> > [1] N. A. Göring et al.  “Out-of-Domain Generalization in Dynamical Systems Reconstruction,” in Proceedings of the 41st International Conference on Machine Learning, PMLR, Jul. 2024, pp. 16071–16114. Accessed: Oct. 07, 2024
> >
> > [2] F. Hess et al. “Generalized Teacher Forcing for Learning Chaotic Dynamics,” in Proceedings of the 40th International Conference on Machine Learning, PMLR, Jul. 2023, pp. 13017–13049.
> >
> > [3] A. S. Krishnapriyan et al., “Characterizing possible failure modes in physics-informed neural networks,” in Advances in Neural Information Processing Systems, vol. 34, 2021.
> >
> > [4] Vlachas et al., “Data-driven forecasting of high-dimensional chaotic systems with long short-term memory networks,” Proceedings of the Royal Society A: Mathematical, Physical and Engineering Sciences, vol. 474, no. 2213, p. 20170844, May 2018
> >
> > [5] Z. Li et al., "Learning chaotic dynamics in dissipative systems," Advances in Neural Information Processing Systems, vol. 35, pp. 16768–16781, 2022.

---

> > > ### Author Response · Authors · 2024-11-29
> > > **any comments?**
> > >
> > > Dear Referee LBGT,
> > >
> > > We wondered whether you would have any more questions or comments on our revision?
> > >
> > > Thanks!

---

> > > > ### Author Response · Authors · 2024-12-03
> > > >
> > > > Dear Referee, even just a brief feedback on how you felt our rebuttal addressed your points would be much appreciated. Thank you!

---

### Author Response · Authors · 2024-11-21
**General reply/ summary of changes**

We thank all four referees for their careful reading of our manuscript and the many great suggestions and valuable points, which we address in more detail in our point-by-point replies below. While three of the referees were very positive, or outright enthusiastic, about our paper, one referee found our work also very interesting, but remained a bit more reserved in general. We hope the additional material and experiments we provide, our substantial rewriting of unclear sections, and the replies we provide below, will also convince this referee!

In summary, besides many textual changes and updates to some figures, we added the following new material to our revision:

1) We tested alternative architectures embedded into our hierarchical framework, namely simple vanilla RNNs, LSTMs, and Markov Neural Operators (MNOs), which include physical priors, to demonstrate the generality of our approach (Appx. A.1.6, A.5.1, new Fig. 17 and Table 3).

2) We tested whether it is also feasible to integrate information from completely *different types of systems* (not only different dynamical regimes from the same type of system) within the same hier-shPLRNN, see Appx. A3/B and new Fig. 27.

3) We provide several analyses that confirm the robustness, performance scaling as function of the number of subjects and time series length, and the computational efficiency of our approach, see newly added last pg. of sect. 4.1, Appx. A 1.5, A.1.8, A.1.9, and new Figs. 11-14.

4) We tested the approach with different observation models, see sect. 3.1 and Appx. 1.6.

5) We tested our framework on another real-world dataset more common in the time series literature, namely electricity load, see Appx. A.3/B and new Fig. 26.

6) We included an illustration of the crucial differences in objectives between DSR models and classical time series models like N-HITS or temporal CNNs, new Fig. 28.

7) We thoroughly reworked in particular the Methods sect. 3, clarifying various aspects of our methodology, its training procedure, and its hyper-parameter settings, and included a new figure (Fig. 8) illustrating the training pipeline.

---

### Meta-Review · Area_Chair_9ush · 2024-12-20

**Metareview:**

This paper addresses the challenge of learning dynamical systems from multiple time-series datasets by introducing a hierarchical version of the shallow PLRNN model. The proposed method incorporates group-level parameters in a shared latent space, allowing for the extraction of underlying dynamics while preserving individual variability. The authors validate their approach on synthetic and real-world datasets, demonstrating improved performance compared to baseline models, along with interpretability that reveals ground-truth dynamics.

One weakness of the paper is the limited exploration of the method's generalization to models beyond the shPLRNN framework, as well as its dependency on combining established training techniques rather than introducing entirely novel methods. Additionally, the clarity of the related work section could be improved by discussing specific alternative approaches, particularly those designed for multi-session neuroscience datasets, and benchmarking against them where feasible.

This paper is a strong candidate for acceptance due to its significant contributions to the field of dynamical systems modeling, particularly in its ability to incorporate group-level information for improved performance and interpretability. The results, supported by extensive experiments and insightful analyses, highlight the method's utility for scientific applications, where uncovering underlying dynamics is of paramount importance.

**Additional Comments On Reviewer Discussion:**

During the rebuttal period, the authors successfully addressed key concerns raised by the reviewers, leading to a positive evaluation of the revised manuscript. Several critical issues were identified and effectively resolved, strengthening the overall contribution of the paper.

One major concern was the applicability of the method beyond the shPLRNN framework. The authors addressed this by conducting additional experiments with other models, including RNN, LSTM, and Markov Neural Operators, demonstrating the broad applicability of their method. This addition significantly enhanced the manuscript. Another concern was whether the approach meaningfully builds on prior work or simply combines existing techniques. The authors clarified the novelty of their approach, particularly in constructing group- and subject-level parameters, and emphasized the interpretable semantic structure of feature spaces, thus underscoring the practical significance of their work.

Scalability with varying time-series lengths was another concern raised by reviewers. The authors added a new section analyzing performance in relation to time-series length and subject number, addressing the scalability and robustness of their approach. Additionally, the reviewers questioned the comparison to other methods, such as SINDy and PINNs. The authors provided a detailed discussion of the limitations of these methods, incorporating physics-informed approaches into their framework and supporting their claims with relevant experiments.

Concerns about the relevance of tractability for real scientific applications were also addressed, with the authors emphasizing the importance of understanding dynamical systems for scientific interpretation and supporting their claims with related literature. Finally, the reviewers expressed concerns about non-stationarity, particularly in biological systems. The authors added experiments simulating time-varying control parameters, demonstrating the model's potential for modeling non-stationary dynamics, which improved the biological interpretability of the model.

In conclusion, the authors' revisions, which included additional experiments, clarifications, and a detailed rebuttal to the reviewers' concerns, significantly improved the manuscript. While one reviewer remained unconvinced, most concerns were effectively addressed, leading to a positive final decision.

---

### Decision · Program_Chairs · 2025-01-22

Accept (Poster)